# How ice grows from premelting films and water droplets

David N. Sibley [1], Pablo Llombart [2,3], Eva G. Noya [2], Andrew J. Archer [1] & Luis G. MacDowell [3✉]

Close to the triple point, the surface of ice is covered by a thin liquid layer (so-called quasi-liquid layer) which crucially impacts growth and melting rates. Experimental probes cannot observe the growth processes below this layer, and classical models of growth by vapor deposition do not account for the formation of premelting films. Here, we develop a mesoscopic model of liquid-film mediated ice growth, and identify the various resulting growth regimes. At low saturation, freezing proceeds by terrace spreading, but the motion of the buried solid is conveyed through the liquid to the outer liquid–vapor interface. At higher saturations water droplets condense, a large crater forms below, and freezing proceeds undetectably beneath the droplet. Our approach is a general framework that naturally models freezing close to three phase coexistence and provides a first principle theory of ice growth and melting which may prove useful in the geosciences.

[1] Department of Mathematical Sciences, Loughborough University, Loughborough LE11 3TU, UK. [2] Instituto de Química Física Rocasolano, CSIC, Calle Serrano 119, Madrid 28006, Spain. [3] Departamento de Química Física (Unidad de I+D+i Asociada al CSIC), Facultad de Ciencias Químicas, Universidad Complutense de Madrid, Madrid 28040, Spain. ✉email: lgmac@quim.ucm.es

The growth and melting of ice play a crucial role in numerous processes, from the precipitation of snowflakes[1], to glacier dynamics[2], scavenging of atmospheric gases[3] or climate change[4]. Yet, despite ice ubiquity both in large masses on the poles and as tiny crystals in the atmosphere, we still do not fully understand how ice actually grows (or melts)[5–8].

Conflicting experimental measurements of ice growth rates[9–13] have been analyzed under a framework of classical crystal growth based on direct deposition from the vapor phase, followed by the subsequent two-dimensional migration of adatoms onto surface kinks[14]. However, the last two decades have witnessed great progress in the experimental characterization of the ice/vapor interface at equilibrium[7]. Results from different experimental techniques[15–18], as well as computer simulations, confirm that the surface disorder of ice grows steadily as the triple point is approached, and what is sometimes referred to as a "quasi-liquid layer" of premelted ice is formed on its surface[19–23]. Unfortunately, classical growth models based on the terrace-ledge scenario do not account for the impact of premelting films at all, and attempts to incorporate this effect have met only limited success[24–26].

Our current understanding of snow crystal growth illustrates this uncomfortable situation. The primary habit or aspect ratio of these familiar hexagonal crystallites can change dramatically with small changes in temperature and saturation, from extremely elongated needle-like crystals to almost flat plate-like dendrites[27]. But despite their variety and complexity, these shapes can be described using phenomenological models with amazing accuracy, based on just a number of parameters[28,29]. Particularly, the primary habit is dictated by a kinetic growth anisotropy factor, describing the ratio of horizontal to vertical growth rates[29]. Unfortunately, the mapping of this phenomenological parameter to the actual ambient conditions in the atmosphere, namely, temperature and water saturation, remains a long-standing topic in crystal growth science[12,13,25]. Accounting explicitly for the premelting layer appears an essential requisite to unveil the dependence of growth rates on ambient conditions.

The difficulty to incorporate the role of premelting films on crystal growth theories is also encountered in many systems of interest in materials science[30–32], where the partially stable liquid phase can even condense into liquid droplets on the growing substrate[15,33–35] and change the mechanism of crystal growth substantially.

The problem is akin to one encountered in the theory of wetting, where one studies how a metastable liquid phase (say, water), adsorbs at the interface between a solid substrate (ice) in contact with a vapor (water vapor) as the liquid/vapor coexistence line is traversed[36]. For an inert substrate, wetting is very well understood in terms of the underlying interface potential $g(h)$ that measures the free energy of the adsorbed film as a function of film thickness $h$[37]. Out of equilibrium, however, the substrate continually feeds from the adsorbed film at the expense of the mother phase, so it is debatable whether it is possible to define meaningfully a film thickness and corresponding interface potential.

Here, we combine state-of-the-art computer simulations, equilibrium wetting theory, and thin-film modeling to describe the kinetics of the ice surface in the vicinity of the triple point within a general framework for wetting on reactive substrates. Our results show that as the vapor saturation increases, ice first grows by terrace spreading below a premelting film with a well-defined stationary thickness. At higher saturations, however, the premelting layer thickness diverges, and growth actually proceeds from below a bulk water phase. In between these two regimes, at intermediate saturations, droplets condense on the ice surface, and growth proceeds mainly under the droplets. The different regimes are separated by well defined kinetic phase lines, whose location can be mapped to an underlying equilibrium interface potential.

## Results

**Interface potential for water on ice**. Most experiments in the literature report premelting layer thicknesses as a function of temperature. However, premelting can also be understood as the condensation of water vapor onto the bulk ice surface. Viewed as an adsorption problem, one sees that the layer thickness is both a function of temperature and vapor pressure[25]. Strictly, ice in contact with water vapor can only be in equilibrium along the sublimation line. It follows that the premelting thickness away from the sublimation line can only be meaningfully characterized for small deviations away from coexistence, where vapor condensation and freezing occur at exactly the same rate. Ice can then be out of equilibrium, while the premelting film remains in a stationary state of constant thickness[38]. The failure to recognize this important point is the source of much confusion in the literature and largely explains why results for the premelting layer thickness differ by orders of magnitude close to the triple point.

Here, we show that an analysis of equilibrium surface fluctuations of ice along the sublimation line can be exploited to calculate an approximate interface potential for the premelting film. Input in a suitable theory of crystal growth dynamics, this allows us to characterize the premelting layer thickness at arbitrary temperature and pressure.

To see this, we write the effective surface free energy per unit surface area at solid/vapor coexistence as $\omega(h;T) = g(h;T) - \Delta p_{lv}(T)h$, where $\Delta p_{lv}(T)$ is the pressure difference between the liquid and vapor bulk phases at the solid/vapor coexistence chemical potential. The free energy $\omega(h;T)$ may be calculated over a limited range of $h$, by simulating at solid/vapor equilibrium. During the course of the simulation, the film thickness fluctuates according to $P(h;T)$, a probability distribution which can be easily collected. This can be used to obtain the free energy from the standard fluctuation formula $A\,\omega(h;T) = -k_B T \ln P(h;T)$, where $k_B$ is Boltzmann's constant and $A$ is the surface area[39,40]. On the other hand, $\Delta p_{lv}(T)$ is a purely bulk property and can be readily calculated by thermodynamic integration from available data (see "Methods" and ref. [41]). With both $\omega(h;T)$ and $\Delta p_{lv}(T)$ at hand, a batch of simulations along the simulation line can provide $g(h;T) = \omega(h;T) + \Delta p_{lv}(T)h$ for a set of temperatures over a range of overlapping film thicknesses. Since the interface potential is expected to exhibit only a small temperature dependence, the set of piecewise functions $g(h;T)$ at different temperatures may be combined to build a master curve $g(h)$ over the whole range of film thicknesses spanned in the temperature interval of the simulations (see "Methods" and Supplementary Note 1).

In principle, computer simulations of ice premelting are extremely challenging. The environment of a given molecule changes from solid to liquid and then to vapor over the scale of just one nanometer or less. The local polarization changes significantly across the interface, and therefore the average many-body forces differ greatly depending on the local position. Such a complicated situation is best described with electronic quantum-mechanical calculations, or explicit many-body potentials[42,43]. Unfortunately, simulations with this level of detail for system sizes as large as required here appear unfeasible. Therefore, we employ the TIP4P/Ice model[44]. Although this is a point-charge non-polarizable potential, it predicts accurately both the solid/liquid and liquid/vapor surface tensions[45]. Furthermore, in the range between 210 and 271 K, it produces film thicknesses that lie between 3 and 10 Å, consistent with a growing body of evidence from experimental probes[17,18,46].

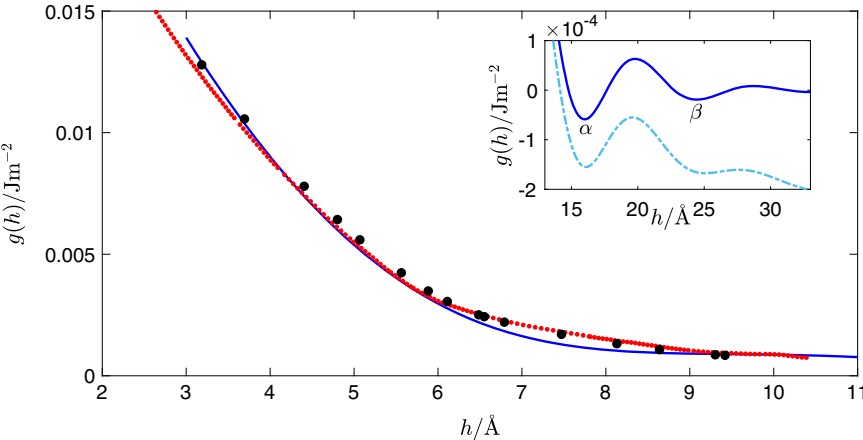

**Fig. 1 Interface potential for a water film adsorbed on ice as calculated from computer simulations.** The small red circles are simulation results obtained from this work. The larger black circles are results obtained by integration of the related disjoining pressure as determined recently[41]. The dark solid blue line is a fit to the simulation results, constrained to exhibit two minima. The inset shows details of the primary $\alpha$ and secondary $\beta$ minima, which are not visible on the scale of the main figure. For an inert substrate, the $\beta$ state is stabilized at pressures $\Delta p = 46{,}000$ Pa above liquid–vapor saturation (dot-dashed light-blue line).

The results obtained with the TIP4P/Ice model for thicknesses up to one nanometer are analyzed as described above to produce the interface potential shown in Fig. 1.

In practice, the equilibrium film thickness can grow well beyond one nanometer as the triple point is approached, so that a complete model of the interface potential requires additional input from theory and experiment.

Mean-field liquid state theory shows that a short-range contribution of the interface potential originating from molecular correlations in the adsorbed film obeys the following equation[47–50]:

$$g_{sr}(h) = C_2 \exp(-\kappa_2 h) - C_1 \exp(-\kappa_1 h) \cos(q_0 h + \alpha), \quad (1)$$

where $C_i$ are positive constants, $\kappa_1$ and $\kappa_2$ are inverse decay lengths (whichever is shorter is the inverse bulk correlation length), and $q_0 \approx 2\pi\, d_0^{-1}$, where $d_0$ is the molecular diameter.

In practice, small amplitude capillary wave fluctuations at both the solid/liquid and liquid/vapor interfaces considerably wash away the oscillatory behavior and renormalize the mean-field coefficients. Our computer simulations for the interface potential of the basal face are consistent with this scenario: fits describe the simulations accurately up to 10 Å, and then predict a fast decay with very weak oscillations of the sinusoidal term (c.f.[41]).

In addition, there are algebraically decaying contributions to the interface potential which stem from the long-range van der Waals interactions. These forces originate from fluctuations of the electromagnetic field. Elbaum and Schick[51] parameterized the dielectric response of ice and water to numerically calculate these contributions with Dzyaloshinskii–Lifshitz–Pitaievsky theory. Following ref. [52], we show that the resulting crossover of retarded to non-retarded interactions is given accurately as

$$g_{vdw}(h) = -Bh^{-3}[1 - f\exp(-ah) - (1-f)\exp(-bh)], \quad (2)$$

where $f$ is a parameter that accounts for the relative weight of infrared and ultraviolet contributions to the van der Waals forces, $a$ is a wavenumber in the ultraviolet region, while $b$ falls in the extreme ultraviolet and accounts for the suppression of high-frequency contributions (see Supplementary Note 2 for further details).

The algebraic decay of the van der Waals forces provides a negative contribution to the interface potential and produces an absolute minimum at finite thickness[41,51]. This explains the observation of water droplets formed on the ice surface just a

few Kelvin away from the triple point[15,34,53]. The droplets observed in the experiment have a small contact angle of $\theta \sim 2°$, which imply a shallow primary minimum with energy $\gamma_{lv}(\cos\theta - 1) \sim -10^{-5}$ J m$^{-2}$.

Combining all this information, we obtain $g(h) = g_{sr}(h) + g_{vdw}(h)$ and fit our computer simulation results to this form, with $C_i$, $\kappa_i$, $q_0$, and $\alpha$ as fit parameters (Supplementary Table 1 and Supplementary Note 3). In fact, the simulation results can be fitted very accurately to $g_{sr}(h)$ alone[41], but the extrapolation of the simulation results to larger $h$ is required to describe the behavior at saturation. Therefore, in the parameter search we impose that $g(h)$ exhibits minima at energies $\sim -10^{-5}$ J m$^{-2}$, as observed in experiment[34]. The constrained fit yields an interface potential in good agreement with the available simulation data—see Fig. 1. Consistent with expectations from renormalization theory, the shallow minima in the interface potential are more widely spaced than one would expect from mean-field theory, located at $h_\alpha = 16.0$ Å and $h_\beta = 24.5$ Å. We refer to these two as the $\alpha$- and $\beta$-minima, respectively, and this interface potential provides a transition between a thin $\alpha$ and a thick $\beta$ film at sufficiently large supersaturation as suggested in experiments of ice premelting in the basal facet[34,53].

**Interface Hamiltonian.** The interface potential is adequate for describing the equilibrium properties of homogeneous films. However, in order to account for droplets like that depicted in Fig. 2 and other such inhomogeneities, we must extend our description. Building on previous work[20,45], we begin by constructing a coarse-grained free energy (effective Hamiltonian) with all the required physics, consisting of a coupled sine-Gordon plus Capillary Wave (SG+CW) Hamiltonian with bulk fields,

$$\Omega = \int \left[ \frac{\gamma_{sl}}{2}(\nabla L_{sl})^2 + \frac{\gamma_{lv}}{2}(\nabla L_{lv})^2 - u\cos(q_z L_{sl}) \right.$$
$$\left. + g(L_{lv} - L_{sl}) - \Delta p_{sl} L_{sl} - \Delta p_{lv} L_{lv} \right] d\mathbf{x}. \quad (3)$$

The first two terms account for the free energy cost to increase the surface area of the solid/liquid and liquid/vapor surfaces in a long-wave approximation, where $L_{sl}$ and $L_{lv}$ are the height profiles of the two interfaces, defined as the distances from the solid–liquid and liquid–vapor interfaces to an arbitrary reference

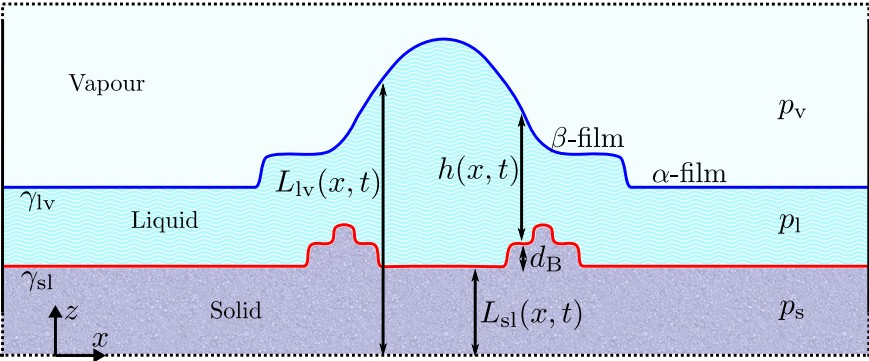

**Fig. 2 Illustration of a possible surface feature with annotations for our two-dimensional gradient dynamics model setup.** Two evolving interfaces are shown: the solid–liquid surface (lower solid red line) at reference height $z = L_{sl}(x, t)$ and above the liquid–vapor interface (upper solid blue line) at reference height $z = L_{lv}(x, t)$. The solid and vapor phases are modeled as extending infinitely below and above, respectively.

plane that is parallel to the plane of the average ice surface (c.f. Fig. 2). Furthermore, $\gamma_{sl}$ and $\gamma_{lv}$ are the solid/liquid interfacial stiffness coefficient and the surface tension, respectively. The cosine term accounts for the energy cost, $u$, to move the solid/liquid surface $L_{sl}$ away from the equilibrium lattice spacing, as dictated by the wave-vector $q_z = 2\pi \, d_B^{-1}$, where $d_B$ is the lattice spacing between ice bilayers at the basal face. This simple model is known to describe adequately nucleated, spiral, and linear growth[54–57]. The interface potential coupling the two surfaces seeks to enforce the equilibrium thickness of the premelting film $h = L_{lv} - L_{sl}$. The last two terms account for the bulk energy of the system as measured relative to the (reservoir) vapor phase with fixed chemical potential $\mu$, where $\Delta p_{sl} = p_s(\mu) - p_l(\mu)$ is the pressure difference between the bulk solid and liquid phases, while $\Delta p_{lv} = p_l(\mu) - p_v(\mu)$ is the pressure difference between the bulk liquid and vapor phases. These two terms account for the free energy change due to growth/melting of the solid phase at the expense of the premelting film, and exchange of matter between the latter and the vapor *via* condensation/evaporation.

Note that the spectrum of equilibrium surface fluctuations of Eq. (3) can be obtained exactly up to Gaussian renormalization[20]. Accordingly, the parameters required in the theory can be obtained in principle by requiring that the spectrum of fluctuations from the theory match the results from experiments or simulations[23,45]. By virtue of this mapping, the input of Eq. (3) is averaged over fluctuations, so that $\Omega$ is to be interpreted as a renormalized free energy, which incorporates consistently all surface fluctuations in the scale of the parallel correlation length.

**Gradient driven dynamics.** The motion of the solid/vapor interface in the presence of a premelting film necessitates us to account explicitly for the different dynamical processes occurring at both the solid/liquid and liquid/vapor surfaces[24–26]. On the one hand, $L_{sl}$ evolves as a result of freezing/melting at the solid/liquid surface, and on the other hand, $L_{lv}$ evolves as a result of both the condensation/evaporation at the liquid/vapor surface and freezing/melting at the solid/liquid surface. Finally, we must also account for advective fluxes of the premelting film over the surface. In practice, since we are concerned only with small deviations away from equilibrium, we can assume the dynamics is mainly driven by free energy gradients with respect to the relevant order parameters[58]. Accordingly, we treat the freezing/melting and condensation/evaporation in terms of non-conserved gradient dynamics, and the advective fluid dynamics of the premelting film using a thin-film (lubrication) approximation,

whence

$$\frac{\partial L_{sl}}{\partial t} = -k_{sl} \frac{\delta \Omega}{\delta L_{sl}} \tag{4a}$$

$$\frac{\partial L_{lv}}{\partial t} = \nabla \cdot \left[ \frac{h^3}{3\eta} \nabla \frac{\delta \Omega}{\delta L_{lv}} \right] - k_{lv} \frac{\delta \Omega}{\delta L_{lv}} + k_{sl} \frac{\Delta \rho}{\rho_l} \frac{\delta \Omega}{\delta L_{sl}} \tag{4b}$$

where $k_{sl}$ and $k_{lv}$ are kinetic growth coefficients that determine the rate of crystallization and condensation at the solid/liquid and liquid/vapor surfaces, respectively, $\eta$ is the viscosity in the liquid film and $\Delta \rho = \rho_s - \rho_l$, where $\rho_s$ and $\rho_l$ are the densities of the solid and liquid, respectively. Models with some similar features were developed in ref. [58–60].

Notice that the deterministic dynamics given by Eq. (4) is driven by the renormalized free energy, Eq. (3). Accordingly, the equation accounts for stochastic fluctuations implicitly, and it may be interpreted as dictating the evolution of the film profiles averaged over all possible random trajectories[61]. Alternatively, replacing the renormalized free energy by a mean-field Hamiltonian, one can assume the above result describes the most likely path of the system[32]. When the fluctuations are small, the coarse-grained Hamiltonian and the renormalized free energy do not differ significantly, and the evolution of the average trajectory becomes the same as the most likely path, as expected in mean-field theory. In Supplementary Note 4, we provide an extended discussion on this issue and show that Eq. (4) may be derived from a fully stochastic-driven dynamics of the mean-field Hamiltonian.

**Kinetic phase diagram.** The time evolution predicted by Eqs. (3) and (4) is extremely rich and varied and the full range can only be obtained numerically. However, if we assume that the surface is on average flat, then we obtain equations that enable us to predict the outcome of the numerical simulations and determine an accurate kinetic phase diagram. Coarse graining the evolution over the time period required to form a single new plane of the crystal, we replace the time derivatives of $L_{sl}$ and $L_{lv}$ by their average values (denoted as $\langle \cdot \rangle$), yielding a rate law for continuous growth (Supplementary Note 5):

$$\langle \partial_t L_{sl} \rangle = \pm k_{sl} \sqrt{\phi_{sl}^2 - w^2} \tag{5a}$$

$$\langle \partial_t L_{lv} \rangle = k_{lv} \phi_{lv} - (\Delta \rho / \rho_l) \langle \partial_t L_{sl} \rangle \tag{5b}$$

where $w = q_z u$, $\phi_{sl} = \Delta p_{sl} - \Pi$, $\phi_{lv} = \Delta p_{lv} + \Pi$ and the disjoining pressure is defined as $\Pi(h) = -\partial_h g(h)$. In Eq. (5a), the plus sign

corresponds to freezing ($\phi_{sl} > 0$), while the minus sign corresponds to sublimation ($\phi_{sl} < 0$).

Despite the coarse graining, Eq. (5) predict a complex dynamics in very good agreement with the numerical solutions of Eq. (4) (see below).

First, for points in the temperature–pressure plane where $\phi_{sl}^2 < w^2$, the crystal surface is pinned by the bulk crystal field and remains smooth. Within this region, continuous growth cannot occur. Instead, the loci of points obeying $\phi_{sl}^2 = w^2$ encloses a region of activated growth, where the crystal front advances *via* nucleation and spread of new terraces[55,56].

For state points where $\phi_{sl}^2 > w^2$, the thermodynamic driving force becomes larger than the pinning field. The surface then undergoes kinetic roughening, and growth can proceed continuously. The growth of the premelting film thickness may be found by subtracting the growth rate of $\langle \partial_t L_{sl} \rangle$ from that of $\langle \partial_t L_{lv} \rangle$, yielding:

$$\left\langle \frac{\partial h}{\partial t} \right\rangle = k_{lv} \phi_{lv} \mp \frac{\rho_s}{\rho_l} k_{sl} \sqrt{\phi_{sl}^2 - w^2}. \quad (6)$$

In practice, we are interested in mapping the phase diagram for quasi-stationary states, where the solid and liquid phases grow at the same rate, so that the premelting film thickness remains constant, i.e., such that $\langle \frac{\partial h}{\partial t} \rangle = 0$[25,26]. Solving for this equality provides a condition for the film thickness as a function of pressure and temperature, which is conveniently written as:

$$\Pi(h) = -\Delta p_k(p_v, T), \quad (7)$$

where $\Pi(h)$ is the disjoining pressure, while $\Delta p_k(p_v, T)$ is a function of the ambient conditions as set by $p_v$, but depends parametrically also on the growth mechanism and rate constants (see Supplementary Note 5).

To illustrate the significance of this equation, consider the simple case of a rough surface, i.e., such that $w = 0$. Then, solving Eq. (6) for stationarity, readily yields Eq. (7), with the kinetic overpressure given in the simple form:

$$\Delta p_k(p_v, T) = \frac{\rho_s k_{sl}}{\rho_s k_{sl} + \rho_l k_{lv}} \Delta p_{sl} - \frac{\rho_l k_{lv}}{\rho_s k_{sl} + \rho_l k_{lv}} \Delta p_{lv}. \quad (8)$$

Notice that $\Delta p_{sl}$ and $\Delta p_{lv}$ are purely bulk quantities that only depend on the imposed thermodynamic conditions of the system, and convey the state-dependent information to the kinetic overpressure (Supplementary Table 2 and Supplementary Note 6). In the limiting case where the substrate is strictly inert, $k_{sl} = 0$, then Eq. (8) becomes $\Pi(h) = -\Delta p_{lv}$ exactly, which is the Derjaguin condition for the equilibrium film thickness on inert substrates. This is very convenient, because we can then predict the outcome of the non-equilibrium dynamics by analogy with the behavior of equilibrium films on inert substrates, albeit with the effective overpressure $\Delta p_k$ replacing $\Delta p_{lv}$. Likewise, one sees that an effective interface potential $\omega_k(h) = g(h) - \Delta p_k h$ determines the dynamics of the system in the quasi-stationary regime.

This allows us to determine the kinetic phase diagram, identifying the regions in $(p, T)$ space where the different outcomes of the interfacial wetting dynamics is to be expected (Fig. 3). In particular, we identify three significant kinetic phase lines:

- The line of kinetic coexistence (dotted-red line in Fig. 3) occurs when $\Delta p_k = 0$. The location of this line can be obtained from Eq. (7), for the choice $\Pi(h) = 0$. States above this line are effectively oversaturated and have stationary film thicknesses consistent with $\Pi(h) < 0$ and are effectively oversaturated. Accordingly, the Laplace condition for droplet formation is met for the first time, and droplets

can be stabilized transiently. However, this occurs well above the liquid–vapor coexistence line, and explains why droplets reported in experiment are formed only above the condensation line[34,53].

- The line of $\alpha \rightarrow \beta$ kinetic transition (dotted-blue line in Fig. 3). At sufficiently high saturation, the linear term in $\omega_k(h)$ stabilizes the $\beta$ state transiently, and it is possible to observe the coexistence between $\alpha$ and $\beta$ states that has been reported in experiments[34,53]. The line where the condition is met is obtained by solving a double tangent construct as in usual wetting phase diagrams (Supplementary Note 5).

- The kinetic spinodal line (dotted-green line in Fig. 3), which occurs when $\Delta p_k = -\Pi_{spin}$, with $\Pi_{spin}$ the value at which the interface potential $g(h)$ predicts that the liquid/vapor interface $L_{lv}$ becomes linearly unstable, i.e., has a spinodal. This condition leads to a line $p_{spin}(T)$ that can be obtained from Eq. (7), for the choice $\Pi = \Pi_{spin}$. Crossing this line signals the region of the p-T plane where ice crystal growth cannot match the rate of vapor condensation, and the premelting film thickness diverges.

The slope of the kinetic coexistence lines is dictated by the ratio of $k_{sl}$ to $k_{lv}$, while the separation between kinetic phase lines is dictated by the depth and free energy separation between the minima.

Using gas kinetic theory, crystal growth theory, and literature data for water and ice, we estimate the model parameters $k_{sl}$, $k_{lv}$, $w$, $\eta$, $\gamma_{sl}$, $\gamma_{lv}$, $\Delta p_{sl}$ and $\Delta p_{lv}$ for the basal surface of ice (Supplementary Table 3 and Supplementary Note 7). These data, combined with the interface potential $g(h)$ from computer simulations, allows us to draw the kinetic phase diagram depicted in Fig. 3. The shaded area surrounding the sublimation line is the region where crystal growth is a slow activated process, only proceeding via step nucleation and growth. In the absence of any impurities to speed up the nucleation, in this regime the substrate is effectively unreactive for time scales smaller than the inverse nucleation rate, and behaves as dictated by the equilibrium interface potential displayed in Fig. 4a. In practice, the experimental systems reported in ref. [34] contain dislocations, so the crystal freezes by spiral growth and the region of unreactive wetting shown in Fig. 3 for the SG + CW model is not observed. The significance of this change in the growth mechanism can be illustrated by setting $w = 0$. In this case, the region of activated growth is removed altogether, growth proceeds continuously, and the kinetic phase lines all meet the solid/liquid coexistence line as they approach the triple point (Supplementary Fig. 1). This regime is also relevant for the prism plane above its roughening transition at about 269 K.

**Interface dynamics**. An extensive set of numerical simulations performed over a wide range of the $p - T$ plane and initial conditions confirms that the outcome of the dynamics is in excellent agreement with expectations from the kinetic phase diagram of Fig. 3. Here, we report results performed for the basal surface at $T = 269.5$ K and varying vapor pressure. Results are reported in reduced units of the model parameters, with $\kappa_1^{-1} \approx 0.49$ nm for the length scale and $\tau = 3\eta (\kappa_1 \gamma_{lv})^{-1} \approx 0.11$ ns for the time scale.

First, we consider a state very near the sublimation line, where the system is found in the region of activated growth, and water vapor freezes by a growth and spread mechanism. The premelting film here is virtually in equilibrium for the time scale of the simulation, and adopts a thickness of roughly two lattice spacings. In our simulations (Fig. 4b–e and Supplementary Movie 1), an initial terrace mimicking a local defect on the solid/liquid surface $L_{sl}$, not observable by optical means, triggers the formation of a

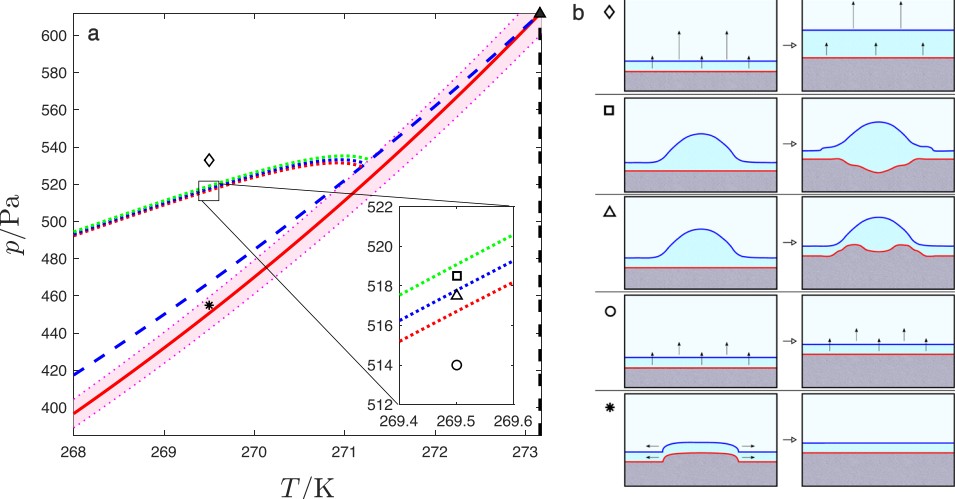

**Fig. 3 Kinetic phase diagram for ice crystal growth.** Panel **a** shows the equilibrium phase diagram and kinetic phase lines. The red solid line is the sublimation line, whereas the dashed lines are metastable prolongations of the vaporization (blue) and melting (black) lines. The filled triangle (▲) indicates the triple point where these lines meet. The remaining features describe the outcome of the dynamics. The shaded area designates the region of activated growth. The dotted lines are kinetic phase lines corresponding to kinetic coexistence (red dotted), kinetic $\alpha \rightarrow \beta$ transition (blue dotted) and kinetic spinodal (green dotted) lines as explained in the text. Panel **b** shows sketches with the dynamics observed in different points of the phase diagram, as indicated with the corresponding symbols. The colored lines describe the ice/liquid (red) and the liquid/vapor (blue) surfaces enclosing the premelting film. The black arrows show the direction of preferential growth. At the point marked by an asterisk (∗), in the region of activated dynamics, growth proceeds by horizontal translation of nucleated terraces. At points such as that marked by a circle (○), above the region of activated dynamics, growth can occur continuously without activation in a steady state of constant film thickness. At points such as that marked by the open triangle (△), above the kinetic coexistence line, droplets can condense and are stabilized transiently with a crater growing inside. At points such as that marked by a square (□), beyond the $\alpha \rightarrow \beta$ line, films in the $\beta$-thick state can be stabilized transiently and form at the rim of the droplet. At higher pressures, past the kinetic spinodal line (green dotted), such as the point marked with a lozenge (◇), the crystal growth rate can no longer match the condensation rate, and the film thickness diverges. The detailed dynamics corresponding to symbols in the phase diagram is illustrated in Figs. 4 and 5.

corresponding terrace on the liquid/vapor surface $L_{lv}$, with a step height equal to the solid lattice spacing. Crystal growth then proceeds by the spreading of the terrace, and the horizontal motion of the solid phase is conveyed to the external liquid/vapor surface. This motion can be observed directly by confocal microscopy, but of course, does not imply the absence of a disordered premelting film (c.f. Fig. 5 in refs. [53] or movie S1 in [34]). Once the new full crystal lattice plane is formed, growth becomes stuck again until a new critical nucleus is formed stochastically.

Crossing the line of nucleated growth toward higher saturation, such that $\phi_{sl} > w$, the thermodynamic driving force is large enough to beat the bulk crystal field, and growth then occurs without activation, as in a kinetically rough surface[54,56]. However, if $\phi_{sl}$ is only marginally larger than $w$, the process occurs in a stepwise fashion, occurring with large time intervals of no growth, followed by height increments equal to the lattice spacing $d_B$ in a short time[26]. On further increasing $\phi_{sl}$, crystal growth then proceeds in a truly quasi-stationary manner while the premelting film thickness remains constant, consistent with Eq. (7).

Interestingly, traversing the metastable prolongation of the liquid–vapor coexistence line does not change the growth behavior in any significant way. Although $\Delta p_{lv}$ is now positive, $\Delta p_k$ is still negative, so the thickening of $h$ is still uphill in the effective free energy $\omega_k(h)$: i.e., the system behaves as if it is effectively undersaturated with respect to liquid–vapor coexistence and the vapor/liquid interface cannot advance faster than the crystal/liquid interface (c.f. Fig. 4f). For a purely flat interface, the stationary film thickness here is therefore somewhat smaller than that found at the sublimation line, but still remains confined within the $\alpha$ state of the interface potential (see Fig. 4f). A liquid droplet quenched to this region of the kinetic phase diagram is never stable – see Fig. 4g–j and Supplementary Movie 2. Instead,

at the contact line of the droplet, terrace formation on the ice is triggered by the action of the disjoining pressure. The crystal then grows and the droplet flattens out, in order to reach a quasi-equilibrium film thickness consistent with Eq. (7). As a transient during the process, the premelting film thickness $h$ can be stable in the $\beta$ film state, reminiscent of the "sunny side up" states observed in experiment[34]. Subsequently, the droplet disappears, leaving an Aztec pyramid-shaped solid surface that is covered by an $\alpha$-thick film. Finally, the inhomogeneity completely disappears, and growth proceeds in a strictly quasi-stationary manner with a flat surface. Notice that during the relaxation process, the droplet is lifted upwards, as a result of the continuous ice growth occurring below. Indeed, comparing Fig. 4g with Fig. 4j, we find that well before the inhomogeneity is washed out, the ice surface grows by about 290 $\kappa_1^{-1}$, at a rate consistent with Eq. (5). This shows that the relevant relaxation time for large inhomogeneities is far larger than the coarse-graining time scale used to obtain the average growth rate law.

The situation changes significantly when saturation is raised above the kinetic liquid–vapor coexistence line, where $\Delta p_k > 0$. For thick enough films, $h$ can now move downhill in the effective surface free energy (Fig. 5a). In this regime, small fluctuations or crystal defects that locally increase the film thickness beyond the spinodal thickness of $g(h)$ trigger the formation of large liquid droplets on top of the premelting film, as observed in experiments —see Fig. 5b–e and Supplementary Movie 3; c.f. Fig. 1-D from ref.[34] Essentially, when $\Delta p_k > 0$, the liquid pressure is large enough to sustain the tension of the droplet surface. However, the droplet cannot be fully stable here, since the system is open. The fastest way to decrease the overall free energy while the solid phase grows is to form a large crater below the droplet and then for the two interfaces to separate. Likewise, a droplet quenched to

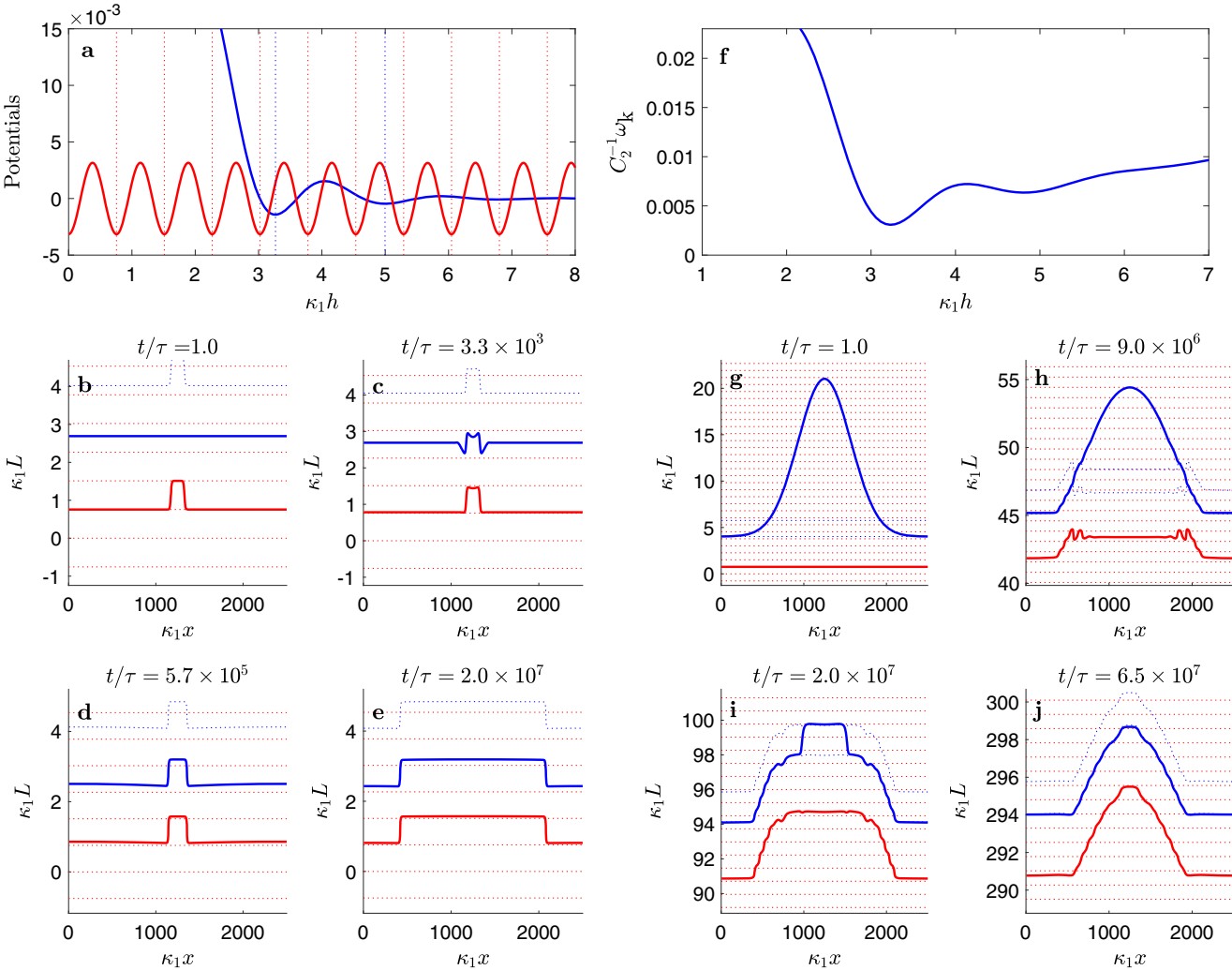

**Fig. 4 Surface dynamics below the kinetic coexistence line.** Panels **a** and **f** show the effective potentials for state points depicted as an asterisk (∗) and a circle (○) in Fig. 3, respectively. Panels **b**–**e** and **g**–**j** show the corresponding solid/liquid and liquid/vapor surfaces at significant milestones in their time evolution (solid red and blue lines, respectively). The dashed red lines indicate the surface location for fully formed ice bilayers, and dashed blue lines show the heights of a premelting film at the α or β minima, as a guide to the eye. Panels **a**–**e** illustrate the evolution in the nucleated regime at $(p, T) = (455$ Pa, 269.5 K) (marked as an asterisk (∗) in Fig. 3). Panel **a** shows the sine-Gordon and interface potential which dictates the surface dynamics. A small terrace nucleated on the solid/liquid surface (panel **b**) triggers the formation of a similar terrace on the liquid/vapor surface (panel **c**) and then spreads horizontally (panels **d**–**e**). Once the surface has flattened, further growth is not possible until a new terrace is nucleated (Supplementary Movie 1). Panels **f**–**j** illustrate the evolution of a droplet quenched to a pressure just below the kinetic liquid–vapor coexistence line at $(p, T) = (514$ Pa, 269.5 K) (shown as a circle (○) in Fig. 3). The effective free energy, $\omega_k(h)$, (panel **f**) inhibits the growth of liquid wetting films. A droplet (panel **g**) triggers the formation of a terrace at the rim, which then spreads inside (panel **h**) and grows to fill the droplet completely (panels **i**–**j**). Subsequent growth occurs in a quasi-stationary state of constant film thickness (Supplementary Movie 2).

this region behaves initially as described above for droplets below the kinetic liquid–vapor coexistence. The difference is that once a few terraces have been formed at the rim, the crystal grows thereon inside the droplet towards its center by creating a premelting film of α thickness, without the droplet curvature flattening out (Supplementary Fig. 2 and Supplementary Movie 5). As growth proceeds, the interface profiles take a transient shape like that of droplets on soft substrates[62,63], with the solid surface growing higher in the contact line region. A crater develops, but is then filled by the growing solid, before the droplet disappears.

Increasing further the pressure above the kinetic α → β transition line, the free energy of the β film becomes less than that of the α film (Fig. 5f). Therefore, a droplet prepared on top of an α film relaxes to a state where it stands on top of the preferred β state. This corresponds to the "sunny side up" configuration found experimentally at sufficiently high saturation—see Fig. 5g–j and Supplementary Movie 4; c.f. Fig. 1-A from ref. [34]. Eventually, the saturation is large enough that the β film metastable minimum is washed away by the linear term $\Delta p_k h$ in $\omega_k(h)$. In this case, the system becomes highly unstable (i.e., linearly unstable to perturbations), and small satellite droplets can form, either in the neighborhood of a larger droplet, or directly from a single local perturbation on the solid surface (Supplementary Fig. 3 and Supplementary Movie 6), a situation that very much resembles experimental observations – see Supplementary Movies S1 and S2 from ref. [34]. Eventually, in the long time limit the inhomogeneities disappear completely, and the premelting film thickness diverges. Crystal growth then proceeds below a macroscopically thick wetting film that feeds on the surrounding bulk vapor.

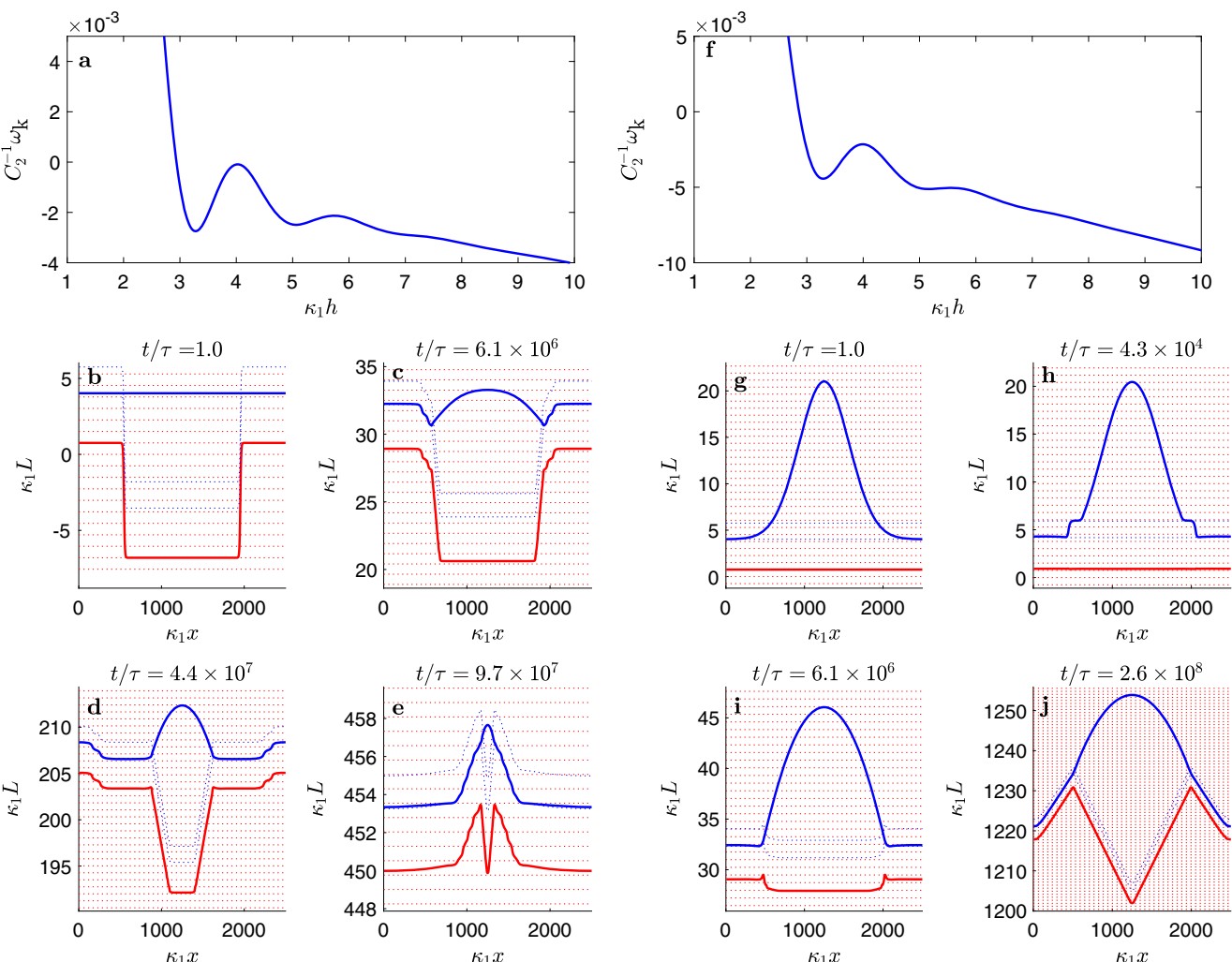

**Fig. 5 Surface dynamics above the kinetic coexistence line.** Panels **a** and **f** show the effective free energies, $\omega_k(h)$ that drive the time evolution of the ice surface at state points depicted as a triangle ($\triangle$) and a square ($\square$) in Fig. 3, respectively. Panels **b–e** and **g–j** show solid/liquid and liquid/vapor surfaces at significant milestones of the dynamics as described in Fig. 4. Panels **a–e** display the evolution of a surface defect at state point $(p, T) = (517.5$ Pa, $269.5$ K) (shown as a triangle ($\triangle$) in Fig. 3). The growth of a thick wetting film is now favorable, as illustrated by the negative slope of the effective free energy in panel **a**. A defect on the solid/liquid surface (panel **b**) triggers the formation of a liquid droplet (panel **c**). Ice then grows inside the droplet, forming a large crater (panels **d–e**) which vanishes eventually when the ice surface catches up with the liquid droplet and attains a stationary premelting layer thickness (Supplementary Movie 3). Panels **f–j** display the evolution of a droplet at $(p, T) = (518.5$ Pa, $269.5$ K) above the kinetic $\alpha \rightarrow \beta$ transition line (shown as a square ($\square$) in Fig. 3). Here, the $\beta$ state has lower free energy than the $\alpha$ state, as illustrated in panel **f**. During the time evolution of a droplet (panel **g**), a thick film of $\beta$ thickness forms at the rim transiently (panel **h**), then the droplet evolves a crater (panels **i–j**) as the ice surface catches up with the droplet (Supplementary Movie 4).

## Discussion

In our study, we have discussed ice premelting, but our results rationalize the behavior of out-of-equilibrium premelting films and wetting on reactive substrates quite generally. In particular, we see that for small deviations away from the sublimation line, freezing occurs in a steady state regime with constant film thickness. In this regime, the thickness of the premelting film is dictated by an equilibrium interface potential and the underlying growth mechanism. For a given growth mechanism, our results show that the outcome of the out of equilibrium dynamics may be predicted accurately from an underlying free energy functional in analogy with wetting on inert substrates. As long as the system remains in this steady-state, the premelting film thickness is well defined and depends both on temperature and pressure.

We emphasize that it is not possible to interpret the dynamics of the quasi-liquid layer without taking into account the behavior of the underlying substrate. In particular, our results demonstrate

that the complex dynamics of a buried solid surface can be conveyed to the experimentally accessible outer surface of the quasi-liquid film. We also confirm that observation of terrace translation, spiral growth, and nucleation observed in experiments is fully consistent with the existence of a nanometer-thick premelting film as observed in simulations[19,20,22,23]. Accordingly, the motion of the experimentally accessible outer surface may be used to interpret the hidden dynamics of the inner surface, very much in agreement with expectations of the Kuroda–Lacmann model[25].

The change from a thin to a thick film regime that occurs across well-defined kinetic lines can result in a significant change in the mechanism for crystal growth. In the thin-film regime, the growth of steps is energetically expensive, because the nuclei are barely buried by the premelting film: steps formed feel a large inhomogeneity as the density changes from solid to vapor across a thin water film. As the kinetic coexistence line is traversed,

however, liquid droplets condense on the ice surface. Steps formed below feel a much smaller tension, similar to that at the ice/water interface. Their free energy of formation is therefore much smaller, and leads to a significant increase in the growth rate at places where droplets have condensed. This has immediate implications for our understanding of ice crystal growth[12,29]. Since crystal corners have high local saturation, droplets are more likely to condense there, providing a source of water for the crystal to feed by growth and spread mechanism from corners towards facet centers as observed in experiments[9,12,64]. Furthermore, small crystallites with large vapor pressure are more likely to have droplets condense at their corners, explaining why the growth mechanism on a basal facet appears to be different in large and in small crystallites[64]. Interestingly, this suggests that droplet condensation could play a role in the tip splitting mechanism of ice grown from the vapor. Advanced optical microscopy appears a candidate technique for the verification of this hypothesis.

In summary, we find a discontinuous change of crystal growth mechanisms with saturation. Combined with recent findings of non-monotonic temperature dependence of step-free energies[23,41], our results could help fill the gap between microscopic theories and mesoscopic models of snowflake growth[29].

## Methods

**Computer simulations.** Simulations of an equilibrated ice slab in the *NVT* ensemble are performed in the temperature range 210–270 K for the TIP4P/Ice model[44] using GROMACS 5.0.5. The equations of motion are integrated using the Leap-Frog algorithm, with a time step of 3 fs. Bond and angle constraints are applied using the LINCS algorithm. The canonical ensemble is sampled using thermostated dynamics with the velocity rescale algorithm[65]. The Lennard–Jones interactions are truncated at a distance of 9 Å. Electrostatic interactions are evaluated using the Particle Mesh Ewald algorithm with the same real space cutoff. We calculate the reciprocal space term using a total of $80 \times 64 \times 160$ vectors in the *x*, *y*, *z* reciprocal directions, respectively. We use a 0.1-nm grid spacing and fourth-order interpolation scheme for the charge structure factor. Simulations are carried out in systems consisting of $8 \times 8 \times 5$ unit cells of pseudo-orthorhombic geometry, each containing 16 molecules. The initial configurations for the solid ice slab are prepared with a random realization of the hydrogen bond network, following ref. [66]. One such initial lattice is provided as Supplementary Data 1. This is then simulated at 1 bar to obtain the equilibrium lattice parameters and placed in vacuum for further equilibration in the *NVT* ensemble during 15 ns. Averages are collected on production runs 35-ns long. During the simulations, we identify structurally liquid-like molecules using the $\bar{q}_6$ order parameter[67]. Once these molecules are identified, we determine the locations of the liquid–vapor and solid–liquid surfaces as explained in ref. [45]. From these two surfaces, we calculate the local film thickness as the difference between these, $h(\mathbf{x}) = L_{\mathrm{lv}}(\mathbf{x}) - L_{\mathrm{sl}}(\mathbf{x})$. For the calculation of the interface potential, the local film thickness for a given configuration is laterally averaged, in order to obtain the average liquid film thickness. The set of global film thicknesses obtained are used to compute the probability histograms $P(h)$, from which $g(h)$ can be calculated as detailed in the Supplementary Note 1. The results for $g(h)$ are fitted to the model described in the main text. Parameter values and further details are given in Supplementary Table 1 and Supplementary Note 3.

**Gradient dynamics.** Numerical computations of the dynamics of the thin-film equations are performed using the method of lines, similar to that used in ref. [68], but with a periodic pseudospectral method for the spatial derivatives. The method is extended to evolve the two interfaces (solid–liquid, and liquid–vapor), with coupling terms involving mass transfer and the two interface potentials naturally included. For the evolution of the solid–liquid interface, a pinning effect in the horizontal direction can occur if too few mesh points are used. Consequently, rather than using an extremely large number of points in the finite difference scheme, we implement a periodic pseudospectral method which significantly increases the rate of numerical convergence. The numerical method uses discretization on a regular (periodic) grid and a band-limited interpolant derived using the discrete Fourier transform and its inverse to form the differentiation matrices which act in real space. The presence of the premelting film avoids the need to explicitly evolve the contact lines, in comparison to some of our previous work using pseudospectral discretisation[69,70]. For the time stepping, the ode15s Matlab variable-step, variable-order solver is used. Our numerical calculations are performed on the non-dimensionalised version of the model equations. We find that choosing $\kappa_1^{-1} \approx 0.49$ nm and $\tau = 3\eta \left(\kappa_1 \gamma_{\mathrm{lv}}\right)^{-1} \approx 0.11$ ns as our units of length and time in the non-dimensionalisation works well. Further details of the

method and initial conditions are given in Supplementary Note 8 and Supplementary References.

**Model parameters.** Phase coexistence data required to compute $\Delta p_{\mathrm{sl}}$, $\Delta p_{\mathrm{lv}}$, structural properties of ice, and surface tension coefficients are obtained from the literature as described in Supplementary Tables 2–3 and Supplementary Note 6. The kinetic growth coefficients $k_{\mathrm{sl}}$ is estimated from the kinetic theory of gases, and $k_{\mathrm{lv}}$ is chosen such that the kinetic coexistence line has a slope similar to experiments. The sine-Gordon coefficient $u = 1.3 \times 10^{-4}$ J m$^{-2}$ is chosen to match step-free energies from the literature. The viscosity is taken from literature values of undercooled water. Further details of the choice of model parameters are given in Supplementary Note 7. The actual model parameters used in this work may be found in Supplementary Tables 1–3.

## Data availability

The data that support the findings of this study are available from the corresponding author upon reasonable request. Source data are provided with this paper.

## Code availability

Numerical simulations for the gradient dynamics in this work are performed using Matlab. All related codes can be built with the instructions provided in "Methods".

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

## Acknowledgements

We are indebted to Uwe Thiele for advice on formulating our gradient dynamics model. L.G.M. is grateful to Loughborough University for hosting a stay funded by the "Programa Estatal de Promoción del Talento y su Empleabilidad en I+D+i" of the Spanish Ministerio de Educación, Cultura y Deporte (Plan Estatal de Investigación Científica y Técnica y de Innovación 2013–2016). We acknowledge the computer resources at MareNostrum and the technical support provided by BSC/MN (QCM-2017-2-0008, QCM-2017-3-0034). P.L., E.G.N., and L.G.M. were funded by the Spanish Agencia Estatal de Investigación under grant FIS2017-89361-C3-2-P and DNS by EPSRC grant EP/R006520/1.

## Author contributions

D.N.S. performed gradient dynamics simulations. P.L., E.G.N., and L.G.M calculated interface potential. D.N.S., A.J.A., and L.G.M constructed the continuum model. L.G.M. designed research. D.N.S., A.J.A., and L.G.M, wrote the paper.

## Competing interests

The authors declare no competing interests.
