## [Peer Review File · Nature Communications]

REVIEWER COMMENTS

Reviewer #1 (Remarks to the Author):

Review of the article NCOMMS-20-09296:

"How ice grows: role of surface liquid films and water droplets"

by:

David N. Sibley, Pablo Llombart, Eva G. Noya, Andrew J. Archer and Luis G. MacDowell

submitted in Nature communication.

In this article, a 1D mesoscopic model for ice growth in supersaturated vapor is developed in the range of temperature (-5°C to 0°C) where the quasi-liquid-layer framework is valid. The present model mixes numerical and analytical methods, in phenomenological perspective. An interface potential controlling the distance between solid-liquid and liquid-vapor interfaces, and thus the width h of the premelted layer is first proposed. It embodies a mean field contribution for short range interactions which notably accounts for molecular packing effects, and a long range interaction contribution, accounting for van der Waals interactions. To parametrize the potential, first, molecular dynamics simulations are performed in the NVT ensemble, using the GROMACS package, at different temperatures between 271 K and 210 K and along to the sublimation line. This provides an effective surface free energy per unit surface area defined piece wise on a small range of h for each temperature. Then, the continuous effective surface free energy density per unit surface is reconstructed using histogram reweighting technique, on the full range of premelted layer width (0-10 Å) addressed by MD simulations. Therefrom, the numerical potential can be estimated from a linear development close to the sublimation line in the phase diagram, for a premelted layer with width comprised between 0 and 10 Å. This represents the first numerical contribution to the model. The second step is the fitting of the theoretical expression of the potential (mean field + van der Waals contribution), to match the numerical interface potential at small premelted layer width, under the constraint of an experimentally known depth for the first well of the potential at larger values of h . This results in a potential with two wells α and β , corresponding to the two equilibrium width of the premelted layer. From that point, an effective grand potential (grand canonique ensemble) for a heterogeneous system is proposed, which encapsulates: the interface potential previously derived, interface energies, Lagrange terms traducing the fixed chemical potential (written in terms of pressure variables), and energy cost to move the solid/liquid surface away from the equilibrium lattice spacing. This provides the salient ingredient of the 2 non conservative phase-field equations governing the dynamics of the system. From these equations, two parallel approaches are proposed. On the one hand, numerical simulations are performed with no further approximation, for about 5 meaningful cases (figure 5). This is the second numerical contribution of the paper. On the other hand, a time averaged version of the equations is proposed. By means of a case-study, different kinetic regimes are identified, resulting in a kinetic phase diagram (figure 4). Shortly put, the main growth regimes are separated by three kinetic transition lines, parametrized by the temperature and the vapour pressure:

-kinetic coexistence line above which the thickness of the premelted layer corresponds to the α well (thinner premelted layer)

-the $\alpha \rightarrow \beta$ kinetic transition line, above which the thicker premelted layer is more favorable (corresponding to β well)

-spinodal line above which the thickness of the premelted layer diverges

-Also emphasized is the nucleated regime close to the sublimation line, with outward propagation of both an ice and a liquid terrace.

Noteworthy, the simulations displayed in figure 5 are spotted on the phase diagram to illustrate the different steady state regimes.

Overall, the present work is very nice and sound. Moreover, albeit most methods at use in the paper are more or less standard, the general approach seems fairly original to me, and results are interest. For this

reason, I recommend the publication in nature communications, provided the authors answer several questions/concerns. In addition, for what it's worth, I personally enjoyed reviewing this piece of work. However, the article+supplementary information couple is very dense, and requires a lot (too much ?) effort to embrace fully. Most of the recommendations hereafter are potential upgrades to improve readability and contextualization of the addressed problem and proposed results .

1/ Motivation, range of the study: The paper provides quite vague motivations for the development of new models (such as the present one) for ice growth rates determination, "precipitation of snowflakes", "glacier dynamics" and so on. I am more concerned about actual application of this work, and more specifically, in the context of multiscale models for ice growth. In particular, this work lies between the applicability range of MD simulations of ice surface dynamics, and phase-field models. Now, phase field models addressing ice dendrite growth are very scarce, especially when it comes to ice growth in vapor [1,2]. Among other reasons, this reflects the lack of relevant description of ice crystal surface growth, that could be used in practice to parametrize phase field models. For that reason, I suggest the authors to add some sentences about this point in the introduction of the paper, and support this discussion by referring to [1,2]. In addition, I would be interested in a more detailed answer (not included in the text) about how relevant information could be extracted from the present model to feed higher scale simulations, which are in desperate need for new description of interface kinetics, even if it is only valid in the QLL regime .

2/ The model is developed without saying a word about one of the most important features of ice growth, especially in vapor : the vertical/horizontal preferential growth, which i believe is connected on the QLL range of temperature to the difference in wettability of prismatic and basal ice facets. This is far from being a detail, as even of this range of temperature, the fast growth direction of ice switches from vertical to horizontal for temperatures close to -3°C . This is obviously connected to the width of the premelted layer, and I cannot see why it could not be at least discussed in the present work .

More generally, at a given temperature, the orientation of the ice surface (prismatic/basal) should be taken into account. This is probably already the case in the present model, but it is not emphasized. As far as I understood, this could be accounted for through the tuning of one or more of the following terms :

- energy cost $u \cdot \cos(\theta) \cdot L$ in the coarse grain grand potential
- van der Waals contribution to the potential
- molecular packing effect in the short range interaction contribution to the potential.

Whatever the answer, this point should be discussed somewhere in the article. For that purpose, maybe the authors could have a glance at Libbrecht's recent publications [3,4]. Also, I believe that the seminal work of Kuroda and Lacmann (ref 26 in the paper), could be of interest for this discussion .

3/Presentation, emphasis and valuation of results: generally speaking, I think that the main achievements of the paper are not presented and described in an optimal way .

-The article climaxes with the kinetic phase diagram (figure 4). First, the figure is obviously too small. Second, it is hard to picture the different premelted layers configuration corresponding to each domain in the diagram. For this reason, figure 4 should be improved by making it bigger first, but also by adding schematic representations of the 5 different premelted layer kinetics selected for the simulations, in a similar way to what was done in figure 1. Putting 1 simulation sample for each case (taken from figure 5) could also do the job. In passing, I am not convinced about the relevancy of figure 1, at least at the beginning of the article, before any explanation. Maybe putting it in the supplementary materials could free some space for a more readable figure 4 .

-The derivation of the limits of the kinetic phase diagram is not clear in the paper: it was impossible for me to understand, without going deep in the supplementary material. In particular, some material from the supplementary information should be brought back in the paper, including equations (39), (41) and (44) + associated explanations, as well as the 3 bullet points p. 18 of the SI. In addition, the explicit connection between the kinetic pressure difference, and the vapor pressure (which is the real parameter in the kinetic

phase diagram) is not transparent. Some words should be added to describe how both connect.

-Finally, regarding numerical results presented in figure 5, the comments in the text and/or markings in the figure should emphasize the most important features, such as the different thickness of the premelted layer corresponding to the alpha or beta wells. Also the validity of the time averaging of kinetic equation (4) should be discussed in light of the duration of transient regimes in the simulations.

4/Generally speaking, the paper lacks comparison with quantitative values, (experimental width of premelted layers etc.). If I am not mistaken, this can only be found at the end of the supplementary materials. I think, the article would improve if more connection with experimental or numerical available data was done.

[1] Demange, G., Zapolsky, H., Patte, R., & Brunel, M. (2017). A phase field model for snow crystal growth in three dimensions. *npj Computational Materials*, 3(1), 1-7.

[2] Demange, G., Zapolsky, H., Patte, R., & Brunel, M. (2017). Growth kinetics and morphology of snowflakes in supersaturated atmosphere using a three-dimensional phase-field model. *Physical Review E*, 96(2), 022803.

[3] Libbrecht, K. G. (2019). Toward a Comprehensive Model of Snow Crystal Growth: 6. Ice Attachment Kinetics near -5 C. arXiv preprint arXiv:1912.03230.

[4] Libbrecht, K. G. (2020). Toward a Comprehensive Model of Snow Crystal Growth: 7. Ice Attachment Kinetics near -2 C (to be published)

Gilles Demange

Reviewer #2 (Remarks to the Author):

In their manuscript MacDowell and coworkers present a computational study on the growth of ice surfaces in the presence of a premelting transition. They combine continuum simulations and theory, based on a generalized Sine-Gordon model, with molecular simulation of a fixed charge model of water, to establish different interface growth regimes. They relate these regimes to corresponding position on the equilibrium phase diagram, and to experimental microscopy studies. While the overall manuscript is interesting, novel, and reasonably written, I have some reservations regarding its conclusions.

Specifically, the conclusions of the manuscript are largely derived from mean field analysis. However, many of the energy scales in their effective Hamiltonian are small relative to kT , calling into question the appropriateness of neglecting fluctuations. This concern is heightened by the lack of corroborating simulation evidence, from molecular or continuum models. Ideally, simulations incorporating fluctuations should be presented at particular points in the phase diagram. Otherwise sharp analytical arguments should be constructed to clarify the robustness of the authors conclusions.

In addition to this broad point, I have a number of more specific concerns.

1. The molecular simulations seem unnecessary. The interfacial potential computed from simulations does not include the primary features invoked for the rich kinetic phase diagram. Unless I am mistaken, the authors fit a damped exponential (Eq. 1) to a monotonically decaying function (red dots in Fig 2) to determine the location of critically important minima h_{α} and h_{β} . There thus seems to be no a priori justification for the locations of the minima based on simulation results. Unless the simulations could be extended to larger h 's the authors should consider dropping the MD results altogether and being more forthright in the construction of a phenomenological model of $g(h)$, albeit one constrained at large length-

scales by experimental data.

2. In the authors wish to keep the calculation of the interfacial potential, they should address concerns I have regarding its calculation. Specifically, the histogram reweighting procedure employed to extract the interfacial potential seems ad hoc, and without theoretical justification. As is well known in multi-canonical sampling or ensemble reweighting, there exists an exact relationship between fluctuating observables at one temperature, and those at another temperature. This relationship depends on the ratio of Boltzmann factors between the two ensembles. In the context of the current study, the statistics of "h" measured in a simulation at T_1 are related to the statistics of h at T_2 through knowledge of the joint distribution of h and E-the energy of the system, reweighted by a factor $\exp(-(1/k_B T_1 - 1/k_B T_2) E + c)$ where c is a normalization constant. It seems the authors are neglecting correlations between h and E, which is not obvious and thus requires justification.

3. It's unclear to me that in Eq. 3, that the oscillatory layering portions of the interfacial potential does not already include contributions from the lattice pinning potential and thus the addition of both does not over-count steric effects.

4. In postulating eq. 4, the authors should more precisely state in terms of their material properties the validity of the lubrication approximation. Are all of their calculations safely within its domain of validity? What sets that scale? If as the authors put it in their introduction the premelting length diverges, its concerning whether their thin film approximation is still valid.

5. In a number of figures captions there are vague comments about the scale of features in $g(h)$. Fig 1 states it arbitrarily increased, and Fig 4 states that is its "too small". These are not given any context in the main manuscript. In Fig 1, this seems deceptive without discussion, in Fig 4 this is out of context and jarring.

6. There is no discussion regarding which facet the authors are considering. Some of their own work point to significant differences in the properties of different ice facets, and so this should be stated along with any generalities that can be drawn.

Reviewer #3 (Remarks to the Author):

This is a very accomplished piece of work and I congratulate the authors. I think this paper could be suitable for nature communications as the model development and application is certainly novel and topical. Although I believe the study is sound and the models valid, I do have some concerns about how the paper is written and a minor technical point.

It's a demanding paper, and feels rather perambulatory in places, particularly in the introduction. I feel it would help the readability of the paper significantly if the authors were to provide a clearer, high level guide to the structure and content of the paper.

In figure 1, it would be helpful if the colour codes for the temperature were identical for both figures.

The MD simulations are based on the TIP4P ice model. The model has a fixed dipole moment. Can the authors comment on how the results may be affected by using a more realistic model, where the dipole is allowed to fluctuate in response to the environment. It is known that the dipole moment of ice varies greatly at the ice surface and of course in liquid water.

It appears all the necessary technical settings to reproduce the work are present although it would be helpful if the authors supplied inputs or at least configuration cells for the MD work.

REVIEWERS' COMMENTS

Reviewer #1 (Remarks to the Author):

The authors fully addressed my concerns of the article. The paper was already very sound and novative, but its density made it very hard to embrace fully. Thanks to the significant amendments of the authors, I am convinced the article can reach a broader community, and it is now easier to grasp the context and target of the paper, as well a extract relevant information.

Therefore, I fully recommend the nice piece of work for publication in nature Communications.

Gilles Demange
Associate Professor
University of Rouen Normandy

Reviewer #3 (Remarks to the Author):

I think the paper has been substantially improved through the revisions all the referees have suggested and I think the paper can be accepted now. The reply to my question has not been dealt with fully. The only way to assess the influence of dipoles is by performing a simulation with a more sophisticated water model such as that due to Paesani et al. but I am not confident such a simulation is tractable and in any case, publication of the study now may encourage the field to strive to test how robust the conclusions from the fixed dipole model are.

Reply to Reviewers

We truly thank the reviewers for the very careful reading of our manuscript and the valuable and constructive suggestions to improve its quality.

We have found some comments often very challenging, requiring us to put in significantly more time and work. We now feel that our understanding of the problem has improved further and responding to the reviewers has allowed us to put our findings in a better perspective.

We also thank very much the referees for their substantial time to read and consider the supplementary material section. Our work combines wisdom from very different communities, including computer simulations, mesoscopic simulations, fluid mechanics, intermolecular forces, surface physics, renormalization and crystal growth theory in a limited space. This makes it a lot to take in, but we also believe that it is the combination of theoretical and numerical tools which has allowed us to make great progress and describe in detail the mechanism of ice growth close to the triple point.

Because of the large number of novel techniques and results employed to characterize ice, and the intricate physics required, we have moved a significant portion of the results into the supplementary material. This has made for a sizable amount of material for the reviewers to consider, but we believe it will very much facilitate easy digestion of the manuscript to most readers willing to trust the high standards of the refereeing process of Nature Communications.

A detailed response to the referees comments and a list of changes is found below.

Reply to Reviewer #1

In this article, a 1D mesoscopic model for ice growth in supersaturated vapor is developed in the range of temperature (-5°C to 0°C) where the quasi-liquid-layer framework is valid. . . Overall, the present work is very nice and sound. Moreover, albeit most methods at use in the paper are more or less standard, the general approach seems fairly original to me, and results are interest. For this reason, I recommend the publication in nature communications, provided the authors answer several questions/concerns. In addition, for what it's worth, I personally enjoyed reviewing this piece of work. However, the article+supplementary information couple is very dense, and requires a lot (too much ?) effort to embrace fully. Most of the recommendations hereafter are potential upgrades to improve readability and contextualization of the addressed problem and proposed results.

We are very grateful for such a careful reading and for the constructive approach of Prof. Demange. We agree that the supplementary material could be the subject of more than one regular paper, and is a somewhat demanding long read for the reviewers. However, we feel that the supporting information is necessary to confirm a number of points that we can only mention briefly in the main text. We expect most readers will trust our peer reviewed supplementary material and will not need to follow the details, but occasionally some will appreciate the supplementary material for additional details when needed.

1-Motivation, range of the study: The paper provides quite vague motivations for the development of new models (such as the present one) for ice growth rates determination, "precipitation of snowflakes", "glacier dynamics" and so on. I am more concerned about actual application of this work, and more specifically, in the context of multiscale models for ice growth. In particular, this work lies between the applicability range of MD simulations of ice surface dynamics, and phase-field models. Now, phase field models addressing ice dendrite growth are very scarce, especially when it comes to ice growth in vapor [1,2]. Among other reasons, this reflects the lack of relevant description of ice crystal surface growth, that could be used in practice to parametrize phase field models. For that reason, I suggest the authors to add some sentences about this point in the introduction of the paper, and support this discussion by referring to [1,2]. In addition, I would be interested in a more detailed answer (not included in the text) about how relevant information could be extracted from the present model to feed higher scale simulations, which are in desperate need for new description of interface kinetics, even if it is only valid in the QLL regime.

In the introduction we have emphasized the generality of our approach to a wide number of problems. We share the reviewer's interest in the the physics of snow crystal growth, and are happy to extend the paper on this issue.

We believe that a number of recent results are allowing to clarify why the growth anisotropy parameter required in phase field models changes with temperature and saturation. The primary habits change mainly due to temperature, and we have recently shown that this occurs as a result of the non monotonic variation of step free energies [1], as suggested long ago by Kuroda

and Lacmann.

For the prism face, a roughening transition occurs at about 269 K. Therefore, in the range from $T=269$ K to 273 K, $w = 0$, and all the faceting disappears. We can illustrate this here for a film simulated under the same conditions as in the paper with all parameters equal but $w = 0$. We find the behavior is similar, but the system relaxes by rounding all edges and terraces. There is indeed a lot of very interesting issues related to this that we could potentially discuss, but we feel our paper is already at the limit of how much to include and to additionally address this issue there would be to include too much.

Figure 1: Dynamics for a system above the roughening transition, with $w = 0$, as could be the case of the prism plane above ca. 270 K. Top: dynamics of a terrace, to be compared with figures 4.a-e. Bottom: Dynamics of a droplet quenched below the kinetic coexistence line, to be compared with figures 4.f-j.

The reason for changes due to saturation remain completely unknown to date, and this paper aims at elucidating this problem. Our kinetic phase diagram shows that different wetting regimes appear discontinuously as a function of saturation, and this can result in the anomalous dependence of growth rates with saturation reported by Libbrecht recently.

Indeed, we have shown that Molecular Dynamics simulations can be exploited to obtain all of the key parameters required including the kinetic growth coefficient [2], the surface tension [3] the interface potential [4] and the step free energies [1, 5]. We are happy to digress on how parameters of our crystal growth model can be extracted from simulation.

The interface potential dictates the thickness of the premelting film, and we have shown recently that the packing correlations which determine the short range behavior of the interface potential make small but significant differences [4]. The van der Waals long range tail could also be different in principle, but the dielectric response of ice is highly isotropic, [6] so we do not expect that the van der Waals interactions, which are the result of electromagnetic fluctuations, will change significantly between facets.

The value of u in the cosine term, together with the stiffness coefficient, dictates the step free energies of the crystal facet and can be obtained from the spectrum of surface fluctuations [4, 5, 7]. Basal and prism facets of ice not only have significantly different values for their

step free energies. We have also shown recently that they have a non monotonic temperature variation [1], very much as suggested by Kuroda and Lacmann many years ago [8]. Plugging these non-monotonic and anisotropic step free energies into our model, provides a first principle theory for ice growth rates that can be used as input into phase field crystal models. We share with the reviewer our excitement on the possibility to explain the Nakaya diagram of ice crystal growth soon with a combination of microscopic and phase field models.

We are happy to extend the discussion on this subject but need to consider also space limitations.

We have added a full new paragraph discussing the current situation of snow crystal modeling in the Introduction. A long paragraph discussing the implications of our work to snow crystal growth has been added in the Discussion.

2-The model is developed without saying a word about one of the most important features of ice growth, especially in vapor : the vertical/horizontal preferential growth, which i believe is connected on the QLL range of temperature to the difference in wettability of prismatic and basal ice facets. This is far from being a detail, as even of this range of temperature, the fast growth direction of ice switches from vertical to horizontal for temperatures close to -3°C . This is obviously connected to the width of the premelted layer, and I cannot see why it could not be at least discussed in the present work.

More generally, at a given temperature, the orientation of the ice surface (prismatic/basal) should be taken into account. This is probably already the case in the present model, but it is not emphasized. As far as I understood, this could be accounted for through the tuning of one or more of the following terms:

-energy cost $u\cos(q*L)$ in the coarse grain grand potential*

-van der Walls contribution to the potential

-molecuar packing effect in the short range interaction contribution to the potential.

Whatever the answer, this point should be discussed somewhere in the article. For that purpose, maybe the authors could have a glance at Libbrecht's recent publications [3,4]. Also, I believe that the seminal work of Kuroda and Lacmann (ref 26 in the paper), could be of interest for this discussion.

In our manuscript we have concentrated on the study of ice growth on the basal facet only. Regrettably this crucial bit of information was missing in the text. Indeed, as the reviewer notices, our theory can be immediately applied to different facets of ice, and, as a matter of fact, to whatever facet of whatever other substance exhibiting premelting.

However, according to our own recent results [4], the change of primary habits with temperature is mainly due to a non-monotonic change of step free energies, that are not given but rather are an input to our theory. Instead, our theory describes how the mechanism of ice growth changes with saturation at constant temperature.

Thanks for pointing out the recent work by Libbrecht. We believe our theory is able to explain the anomalous dependence with saturation reported by him recently. However, we do not think such anomalies are really a crystal size effect. Instead, they are a water saturation effect. Of

course, small crystallites have a higher vapor pressure, but we feel that the results reported by Libbrecht can be explained without considering the crystal size. This is now explained in the Discussion section.

We have now substantially enlarged the discussion and the scope of our model. We describe how the parameters of our theory can be extracted from molecular dynamics simulations, and discuss how small changes in the step free energies can completely change the preferential growth of ice crystals.

3.1- Presentation, emphasis and valuation of results: generally speaking, I think that the main achievements of the paper are not presented and described in an optimal way. -The article climaxes with the kinetic phase diagram (figure 4). First, the figure is obviously too small. Second, it is hard to picture the different premelted layers configuration corresponding to each domain in the diagram. For this reason, figure 4 should be improved by making it bigger first, but also by adding schematic representations of the 5 different premelted layer kinetics selected for the simulations, in a similar way to what was done in figure 1. Putting 1 simulations sample for each case (taken from figure 5) could also do the job. In passing, I am not convinced about the relevancy of figure 1, at least at the beginning of the article, before any explanation. Maybe putting it in the supplementary materials could free some space for a more readable figure 4.

Thanks for this suggestion. We agree that Fig.4 could be improved.

In view of the concerns raised by the three reviewers, we have moved Figure 1 into the supplementary material and largely modified former Figure 4 (now Figure 3) with sketches of significant milestones in the dynamics.

3.2-The derivation of the limits of the kinetic phase diagram is not clear in the paper: it was impossible for me to understand, without going deep in the supplementary material. In particular, some material from the supplementary information should be brought back in the paper, including equations (39), (41) and (44) + associated explanations, as well as the 3 bullet points p. 18 of the SI. In addition, the explicit connection between the kinetic pressure difference, and the vapor pressure (which is the real parameter in the kinetic phase diagram) is not transparent. Some words should be added to describe how both connect.

We are happy to move these explanations into the main text.

The derivation of the kinetic phase diagram is significantly improved with additional discussion and details. Several equations from the supplementary material have now been moved to the main text as suggested by the reviewer.

3.3-Finally, regarding numerical results presented in figure 5, the comments in the text and/or markings in the figure should emphasize the most important features, such as the different thickness of the premelted layer corresponding to the alpha or beta wells. Also the validity of the time averaging of kinetic equation (4) should be discussed in light of the duration of transient regimes in the simulations.

Thanks again for this helpful suggestion. The validity of the time averaging and average growth rate is surprisingly robust. In all cases where stationary growth has set in we can see how the flat regions of the solid/liquid film profile grow stepwise with a rate dictated precisely by this equation. This implies that the transient regimes for equilibration and dissipation of the droplets are many more times larger than those required to average over the step growth (bear in mind the logarithmic time scale of our simulations). This will only break down very close to the nucleated growth line, where the time for growing a step becomes infinite as can be seen from the discussion in the *Supplementary Note 5*.

We have improved the explanation of former Figure 5 (currently Figure 4), and discussed the validity of the growth law in section Interface dynamics.

4-Generally speaking, the paper lacks comparison with quantitative values, (experimental width of premelted layers etc.). If I am not mistaken, this can only be found at the end of the supplementary materials. I think, the article would improve if more connection with experimental or numerical available data was done.

All molecular simulations for all accepted water models predict consistently film thicknesses of subnanometer size up to 2 K away from the melting point. Several experimental studies are also building evidence that this is the case, with significant differences only very close to the triple point [9–12].

Also note that in the section 'Results for the interface dynamics', we match each of our numerical simulations to particular realizations found in experiments.

We have added several new sentences in the manuscript to discuss the consistency of our results with known results from molecular simulation and experiments.

[1] Demange, G., Zapolsky, H., Patte, R., & Brunel, M. (2017). A phase field model for snow crystal growth in three dimensions. *npj Computational Materials*, 3(1), 1-7.

[2] Demange, G., Zapolsky, H., Patte, R., & Brunel, M. (2017). Growth kinetics and morphology of snowflakes in supersaturated atmosphere using a three-dimensional phase-field model. *Physical Review E*, 96(2), 022803.

[3] Libbrecht, K. G. (2019). Toward a Comprehensive Model of Snow Crystal Growth: 6. Ice Attachment Kinetics near -5 C. *arXiv preprint arXiv:1912.03230*.

[4] Libbrecht, K. G. (2020). Toward a Comprehensive Model of Snow Crystal Growth: 7. Ice Attachment Kinetics near -2 C (to be published)

Reviewer #2

In their manuscript MacDowell and coworkers present a computational study on the growth of ice surfaces in the presence of a premelting transition. They combine continuum simulations and theory, based on a generalized Sine-Gordon model, with molecular simulation of a fixed charge model of water, to establish different interface growth regimes. They relate these regimes to corresponding position on the equilibrium phase diagram, and to experimental microscopy studies. While the overall manuscript is interesting, novel, and reasonably written, I have some reservations regarding its conclusions.

Specifically, the conclusions of the manuscript are largely derived from mean field analysis. However, many of the energy scales in their effective Hamiltonian are small relative to kT , calling into question the appropriateness of neglecting fluctuations. This concern is heighten by the lack of corroborating simulation evidence, from molecular or continuum models. Ideally, simulations incorporating fluctuations should be presented at particular points in the phase diagram. Otherwise sharp analytical arguments should be constructed to clarify the robustness of the authors conclusions.

We thank Reviewer #2 for the careful reading of our revised manuscript and his suggestions.

Thin film hydrodynamic models of the type used in our manuscript have for some time now been used to describe accurately the complex dynamics of films of liquids on surfaces. For example, in the abstract of Ref. [13] the authors write Here we demonstrate, for the first time, that the full complex spatial and temporal evolution of the rupture of ultra-thin films can be modelled in quantitative agreement with experiment. The effect of thermal fluctuations can be incorporated, as shown in Ref. [14–16], which results in a stochastic generalization of the thin film equation. The type of situation where thermal fluctuations matter the most is where the liquid wants to dewet from the surface, where the fluctuations can influence the characteristic time-scales of the dewetting process of linearly unstable thin films. A recent paper compares the stochastic and deterministic dynamics with MD simulations, and shows that, whereas a very accurate description is provided by the stochastic equation, the qualitative features of the relaxation process remain unchanged [17]. This is in agreement with the overall picture emerging from the literature, namely, that for low Reynolds number thin film surface flows where the contact angle is not too big, the thin film equation is accurate [14, 16, 18]. These conditions apply for the situations we consider in our work, so we are confident that our model is at least qualitatively correct. For these types of systems, adding fluctuating terms to the equations essentially just changes the time it takes the system evolve through the underlying free energy landscape, but it does not change the underlying landscape itself. This is because the low Reynolds number dynamics is effectively over-damped. A similar situation arises when one considers the dynamics of interacting colloids, where one can derive fluctuating dynamical equations, but these almost always just change the rate at which the system evolves through the underlying free energy landscape [19].

Also notice that in equilibrium, thermal fluctuations could affect our results in two ways. Firstly, at the ice/water surface, thermal fluctuations can transform a faceted surface into a rough

surface across a roughening transition [20]. However, in our paper we deal with the basal surface of ice, which remains smooth up to the triple point [8]. This means that thermal fluctuations are unable to beat the bulk crystal field at the ice surface. Secondly, thermal fluctuations can renormalize the interface potential. However, in this system the interface potential has an algebraic decay that is dominated by long range van der Waals forces. In the presence of long range forces, it is well known that thermal fluctuations are also unable to drive a wetting transition and do not change the mean field scenario significantly [21]. The only concern then is whether the transition from the first to the second minimum of the potential could be washed out by renormalization. However, our fit to molecular dynamics simulations, which is renormalized on the scale of the lateral dimensions ca. 50 nm^2 supports the two minima scenario even after renormalization, and so do experiments [22, 23].

In order to show our interface potential is adequately renormalized, we have exploited the data of our recent paper [1], where we simulated the basal surface over a very large system size with lateral area 245 nm^2 and recalculated the interface potential. The figure below shows that there is at most a very weak system size dependence, hardly observable on the scale of the simulations which confirms the absence of diverging correlation lengths.

Figure 2: Interface potential as reported in this work (black line), compared with new calculations with the same method for a system size 5 times larger (symbols). Right, comparison of the piecewise functions for the small (lines) and large systems (symbols).

Be as it may, we believe that the most salient feature of the model is the presence of one important primary minimum that is the result of opposing short range and long range forces, and this is definitively confirmed in our calculations.

Finally notice that in our paper we are mainly concerned with the time evolution of dissipative systems under external forcing, where the system is driven by strong bulk fields which inevitably lead to crystal growth. In this situation the role of fluctuations might be even less significant than under thermal equilibrium. Yet, one could then interpret that a deterministic equation as the one used by us is describing the most likely path out of all the possible random trajectories, in line with similar interpretations in dynamical density functional theory. [24, 25]

However, we stress that our deterministic equations do not feed from a mean field Hamiltonian, but from an accurate free energy functional that has been renormalized by thermal capillary waves on the scale of our molecular dynamics simulations. Accordingly, the deterministic equations can be interpreted as describing the evolution of the film profiles averaged over the ensemble of all trajectories consistent with the initial conditions [19, 26].

We have now formulated a full stochastic theory that generalizes the stochastic thin film equation for crystal growth and premelting films. This is described at length in new Supplementary Note 4 of the supplementary material. We there show that a deterministic equation based on a renormalized free energy functional as that used by ourselves can be considered as describing the time evolution of the system averaged over an ensemble of trajectories.

1-The molecular simulations seem unnecessary. The interfacial potential computed from simulations does not include the primary features invoked for the rich kinetic phase diagram. Unless I am mistaken, the authors fit a damped exponential (Eq. 1) to a monotonically decaying function (red dots in Fig 2) to determine the location of critically important minima h_α and h_β . There thus seems to be no a priori justification for the locations of the minima based on simulation results. Unless the simulations could be extended to larger h 's the authors should consider dropping the MD results altogether and being more forthright in the construction of a phenomenological model of $g(h)$, albeit one constrained at large length-scales by experimental data.

Notice that all previous efforts to describe the dynamics of premelting film have had to do without the use of an interface potential, which is a crucial thermodynamic input in theories of premelting that has been lacking up to date. [8, 27–29]

Building on our previous work [7, 30–37] here we provide the first quantitative model of the interface potential relevant to ice premelting. The model is constraint using computer simulations at short range and experimental results at long range.

At short range, the essential input into to our method are the distribution of film thickness obtained from simulation, which eventually dictate the average film thickness. Results for film thickness from experimental sources over the last three decades vary over two to three orders of magnitude, c.f. Ref. [38], and it is only with the support from simulations in the last decade that we have been able to constraint this widely scattered data, and critically select a significant number of references that converge with simulation. So the simulation data are indeed very valuable and if we failed to identify this it would leave us wondering which order of magnitude to choose for the film thickness.

Further note that the fit to Eq. 1 does not necessarily produce oscillations. When the purely exponential term decays at slower rate than the damped oscillatory term, the oscillations can vanish. Indeed, for the prism facet we have found that a fit of the interface potential to Eq. 1 produces a purely monotonic function. We attach a Figure from Ref. [4] for the reviewers convenience.

Figure 3: Disjoining pressures for the basal (left,top) and prism (left,bottom) facets, showing fits to the short range oscillatory model of Eq. 1. The corresponding interface potential at large distances is shown in the right. For the basal facet, the fit predicts oscillatory decay; For the prism facet, the decay is monotonic and no oscillations are present. i.e. minima of the interface potential are not input ad-hoc into the model of Eq. 1. Results from Ref. [4]. Note that these fits did not include the van der Waals long range tail.

We take the referees point that we are unable to get the full interface potential $g(h)$ from the simulations. However, as pointed out, the simulations do constrain significantly the overall form and set the key energy-scales contained in the $g(h)$ that we use. We think that if we were to follow the referees suggestion to completely remove the simulation results from the paper (a suggestion that we very seriously considered), future readers of our paper would miss seeing these important aspects and the agreement we do have with simulations (albeit over a limited range of h).

As often is the case, computer simulations only provides limited insight. To grasp the full physics of the problem, we require additional input from theory. The physics of van der Waals forces dictates that the interface potential must have a long range negative decay, but this is often overlooked in the literature. Our simulations definitively confirm a strong short range positive decay of the interface potential. So we can definitively confirm that ice premelting must exhibit one minimum at short range, and the combination of our short range interface

potential and Lifshitz theory dictates that such minimum occurs in the nanometer scale.

Our results further show that the expected analytical form of the renormalized short range interface potential is consistent with both our short range results from computer simulation and the presence of one additional minimum that has been observed in experiments.

We feel that the consistency of our molecular dynamics simulations and experimental findings by a combination of the theory of van der Waals forces, renormalization theory and liquid state theory is a significant achievement that deserves attention.

We have fully rewritten this section to stress the significance of the interface potential and to explain how the simulations constraint the results. Technical details have been moved into the Methods section.

2-In the authors wish to keep the calculation of the interfacial potential, the should address concerns I have regarding its calculation. Specifically, the histogram reweighting procedure employed to extract the interfacial potential seems ad hoc, and without theoretical justification. As is well known in multi-canonical sampling or ensemble reweighting, there exists an exact relationship between fluctuating observables at one temperature, and those at another temperature. This relationship depends on the ratio of Boltzmann factors between the two ensembles. In the context of the current study, the statistics of "h" measured in a simulations at T_1 are related to the statistics of h at T_2 through knowledge of the joint distribution of h and E -the energy of the system, reweighted by a factor $\exp(-(1/k_B T_1 - 1/k_B T_2)E + c)$ where c is a normalization constant. It seems the authors are neglecting correlations between h and E , which is not obvious and thus requires justification.

We agree that the explanation provided for our calculation of the interface potential might have been too short. However, notice that *Supplementary Note 1* provides an in depth explanation and justification of the method.

We also agree that our use of 'reweighting' might have been confusing. In the text we used 'reweighting' to mean that the histograms are reweighted by a factor $\exp(-A\Delta p h/k_B T)$. We agree this was misleading without further clarification, since this term is often reserved for canonical reweighting as explained by the reviewer.

Finally, note that the multiplier Δp is calculated from the thermodynamic relation

$$dp = \rho s dT + \rho d\mu \quad (1)$$

where ρ is the number density and s is the entropy per particle. This is integrated for a path along the sublimation line where both T and μ change simultaneously. Accordingly, the reweighting factor $\exp(-A\Delta p h/k_B T)$ does account for temperature changes, albeit, admittedly, in a mean field sense as noted by the Reviewer. This is explained in the *Supplementary Note 1*. In practice, the histograms at each temperature provide information on the corresponding interface potential at that temperature only. By sticking together the piecewise functions, we are assuming that the interface potential exhibits a small temperature dependence. This appears very reasonable since both the solid substrate changes and the premelting film correspond

to condensed phases with small changes with temperature.

As noted by the Reviewer, one could in principle calculate the exact interface potential by canonical reweighting. However, this in fact turns out to be numerically unfeasible. We seek for a reweighting of the film thickness distribution, but one cannot do without the underlying bulk solid. Therefore, the energy fluctuations carry information on the bulk solid fluctuations, as well as energy fluctuations from the premelting film. Therefore, the reweighting conceptually poses some difficulties. In practice we checked that it produces exponential factors that issue overflow errors in double precision as soon as the extrapolation is performed beyond 10 K.

The alternative is to produce reweighting of the premelting film energies only, but then the number of particles in the premelting film fluctuates. Accordingly, one would require reweighting over the grand-canonical distribution, which can only be carried out with knowledge of the ice/vapor chemical potential at coexistence. This also poses currently unsurmountable problems. The chemical potential of the solid phase cannot be calculated by any insertion method, while the vapor pressure is so small that it can also not allow for the calculation of its chemical potential.

In view of these difficulties, we find that the approximate interface potential used here is a good compromise. Let us point out a number of properties that our interface potential fulfils, which make it sufficiently physically constraint for our purposes:

- The extremal of the free energy $w(h) = g(h) - \Delta ph$ yields exactly the equilibrium film thickness for each temperature along the sublimation line by construction.
- The interface potential obtained here agrees accurately with estimates from an independent method introduced by ourselves recently [4].
- The portions of the interface potential yield a continuous function with no sign of discontinuity of the derivative.

We have fully rewritten the section on interface potentials in order to provide a somewhat more detailed explanation of the method and clarify its approximate nature. Further details are given in the Methods section. A full explanation is provided in Supplementary Note 1.

3-Its unclear to me that in Eq. 3, that the oscillatory layering portions of the interfacial potential does not already include contributions from the lattice pinning potential and thus the addition of both does not over-count steric effects.

We recently published a separate study on determining just the form of the binding potential see our recent PRL for details [4]. This work shows the interface potential is indeed oscillatory.

The cosine term of the lattice pinning potential is required to have the solid/liquid surface to grow in stepwise fashion by amounts consistent with the underlying bulk lattice. Such

correlations are conveyed into the liquid phase, but decay exponentially due to the isotropy of the liquid phase. Accordingly, we need a cosine term to pin the solid/liquid surface, and a damped cosine term to describe the packing effects conveyed by the solid to the liquid/vapor surface as expected from liquid state theory [39–42].

One way to see that there can be oscillatory contributions in both $g(h)$ and the solid-liquid interface pinning term [i.e. the Sine-Gordon term $u \cos(q_z L_{sl})$], is that one can distinguish the two contributions by first considering the case when the two interfaces are on average flat and the distance between them is large, i.e. $(L_{lv} - L_{sl})$ is large. Then the excess grand potential per unit area is:

$$\frac{(\Omega - \Omega_0)}{A} = u \cos(q_z L_{sl}) - \Delta p_{sl} L_{sl} - \Delta p_{lv} L_{lv} + \gamma_{sl} + \gamma_{lv} \quad (2)$$

i.e. there is still the oscillatory potential acting on the solid-liquid interface, that must be there for the growth of the crystal under an infinitely thick liquid layer to grow step-wise (as it does).

Then, secondly consider the case when the two interfaces approach each other, i.e. when $(L_{lv} - L_{sl})$ is no longer large. Then there is the additional contribution in the above equation from the interface potential:

$$\frac{(\Omega - \Omega_0)}{A} = u \cos(q_z L_{sl}) - \Delta p_{sl} L_{sl} - \Delta p_{lv} L_{lv} + \gamma_{sl} + \gamma_{lv} + g(L_{lv} - L_{sl}) \quad (3)$$

When the solid-liquid interface does not move ($L_{sl} = \text{constant}$) then the Sine-Gordon term does not contribute (change). But the packing correlations stemming from the fixed flat substrate have a Fourier mode with period $2\pi/\sigma$, with σ close to the molecular diameter. Therefore, there must be a damped oscillatory contribution to $g(h)$, which is exactly what the simulation results show.

Added to this that $g(h)$ must have multiple minima to make sense of the experimentally observed terrace on “sunny side-up egg droplets. Therefore, we can be confident that both contain oscillations and we do not “over-count steric effects.

4-In postulating eq. 4, the authors should more precisely state in terms of their material properties the validity of the lubrication approximation. Are all of their calculations safely within its domain of validity? What sets that scale? If as the authors put it in their introduction the premelting length diverges, its concerning whether their thin film approximation is still valid.

The lubrication approximation can be derived exactly from the Navier Stokes equation by assuming low Reynolds number and that the vertical variations in the film thickness are small compared to the horizontal variations on the plane of the surface [43, 44]. However, this name is perhaps misleading, because much subsequent work has shown that as long as contact angles remain less than 45° , the film remains sub-millimeter in thickness and the Reynolds number is low, then the thin film equation is highly accurate. This can be understood when one considers

the alternative gradient dynamics formulation for deriving the thin-film equation [45]. The dynamics we consider here is for liquid films with thickness tens of nanometers or less, contact angles that are much less than 10° and have an extremely low Reynolds number. We are very confident that the thin film approximation is valid!

Even when the premelting length diverges, it is still in practice sub-millimeter, and so the approximations are still valid.

5-In a number of figures captions there are vague comments about the scale of features in $g(h)$. Fig 1 states it arbitrarily increased, and Fig 4 states that is its "too small". These are not given any context in the main manuscript. In Fig 1, this seems deceptive without discussion, in Fig 4 this is out of context and jarring.

We agree with this comments, and thank the reviewer for pointing this.

We have moved Figure 1 to the supplementary material and enlarged the caption with further details. We have fully rearranged Figure 4, and very much enlarged the captions. A comment on the role of energy scales in the separation between phase lines is now added in section 'Kinetic phase diagram' and in the Supplementary Note 3.

6-There is no discussion regarding which facet the authors are considering. Some of their own work point to significant differences in the properties of different ice facets, and so this should be stated along with any generalities that can be drawn.

We agree. Surprisingly we missed mentioning that all our results refer to the basal face of ice.

We have fixed this problem and added a discussion on how the model could be applied to the study of other ice facets.

Reviewer #3

This is a very accomplished piece of work and I congratulate the authors. I think this paper could be suitable for nature communications as the model development and application is certainly novel and topical. Although I believe the study is sound and the models valid, I do have some concerns about how the paper is written and a minor technical point.

We thank Reviewer#3 for the comments.

It's a demanding paper, and feels rather perambulatory in places, particularly in the introduction. I feel it would help the readability of the paper significantly if the authors were to provide a clearer, high level guide to the structure and content of the paper.

In the introduction we emphasized the generality of our approach and its potential applicability to a wide number of problems, since we feel Nature Communications seeks a wide audience on all fields of research. However, we do acknowledge this can make the introduction somewhat too abstract. In response to both Reviewer #1 and Reviewer #3 we have added a new paragraph in the introduction that hopefully helps to set the scene for one of the most attractive applications of our work.

We would be happy to add an extensive high level guide to our paper in the introduction, but replying to all queries does not allow us to devote more space to explanations, unfortunately.

We have substantially changed the introduction to make it more clear, and added a high level guide to the paper.

In figure 1, it would be helpful if the colour codes for the temperature were identical for both figures.

This figure is obviously problematic, as it was criticized by all three reviewers.

We have now moved the figure to the supplementary material section, but use a consistent color code for all figures.

The MD simulations are based on the TIP4P ice model. The model has a fixed dipole moment. Can the authors comment on how the results may be affected by using a more realistic model, where the dipole is allowed to fluctuate in response to the environment. It is known that the dipole moment of ice varies greatly at the ice surface and of course in liquid water.

This is a very interesting point. Indeed the dipole increases very much in condensed phases. However, we use a model that has been parameterized to predict accurately properties of condensed water, and particularly, those of ice. Accordingly, the dipole moment is set in fact much larger than it is for an isolated molecule. This is a mess as far as the properties of the vapor phase is concerned, but results in little practical significance, because the vapor phase has an extremely low density (i.e. a vapor pressure just 0.06 atmospheres at the triple point, and much lower at lower temperatures). In principle, the polarizability problem could be a serious

issue at the interfaces, where the effective dipole changes abruptly. In practice, the TIP4P/Ice model (for which one of us contributed to the MC code, the initial configuration and the fitting method, by the way) does impressively well at predicting the ice/water and water/vapor surface tension [1, 5].

We have added a sentence to discuss the role of polarizability and the surface properties of the TIP4P/Ice model.

It appears all the necessary technical settings to reproduce the work are present although it would be helpful if the authors supplied inputs or at least configuration cells for the MD work.

We realize that there is plenty of space in the Methods section, so we have moved some of the technical details that were previously in the supplementary material into the main manuscript.

We acknowledge preparing the initial configuration is not trivial as it requires sampling over random arrangements of the hydrogen bond network.

An initial configuration for our simulations is added as part of the supplementary material.

References

- [1] Llombart, P., Noya, E. G. & MacDowell, L. G. Surface phase transitions and crystal habits of ice in the atmosphere. *Sci. Adv.* **6** (2020).
- [2] Benet, J., MacDowell, L. G. & Sanz, E. Computer simulation study of surface wave dynamics at the crystal–melt interface. *J. Chem. Phys.* **141**, 034701 (2014).
- [3] Benet, J., MacDowell, L. G. & Sanz, E. A study of the ice–water interface using the tip4p/2005 water model. *Phys. Chem. Chem. Phys.* **16**, 22159–22166 (2014).
- [4] Llombart, P., Noya, E. G., Sibley, D. N., Archer, A. J. & MacDowell, L. G. Rounded layering transitions on the surface of ice. *Phys. Rev. Lett.* **124**, 065702 (2020).
- [5] Benet, J., Llombart, P., Sanz, E. & MacDowell, L. G. Structure and fluctuations of the premelted liquid film of ice at the triple point. *Mol. Phys.* **117**, 2846–2864 (2019).
- [6] MacDowell, L. G. & Vega, C. Dielectric constant of ice *ih* and ice *v*: A computer simulation study. *J. Phys. Chem. B* **114**, 6089–6098 (2010).
- [7] MacDowell, L. G., Benet, J. & Katcho, N. A. Capillary fluctuations and film–height–dependent surface tension of an adsorbed liquid film. *Phys. Rev. Lett.* **111**, 047802 (2013).
- [8] Kuroda, T. & Lacmann, R. Growth kinetics of ice from the vapour phase and its growth forms. *J. Cryst. Growth* **56**, 189–205 (1982).
- [9] Bluhm, H., Ogletree, D. F., Fadley, C. S., Hussain, Z. & Salmeron, M. The premelting of ice studied with photoelectron spectroscopy. *J. Phys.: Condens. Matter* **14**, L227–L233 (2002).
- [10] Sadtchenko, V. & Ewing, G. E. Interfacial melting of thin ice films: An infrared study. *J. Chem. Phys.* **116**, 4686–4697 (2002).
- [11] Gelman Constantin, J., Gianetti, M. M., Longinotti, M. P. & Corti, H. R. The quasi-liquid layer of ice revisited: the role of temperature gradients and tip chemistry in AFM studies. *Atmos. Chem. Phys.* **18**, 14965–14978 (2018).
- [12] Mitsui, T. & Aoki, K. Fluctuation spectroscopy of surface melting of ice with and without impurities. *Phys. Rev. E* **99**, 010801 (2019).
- [13] Becker, R. *et al.* Complex dewetting scenarios captured by thin-film models. *Nature Materials* **2**, 59–63 (2003).
- [14] Davidovitch, B., Moro, E. & Stone, H. A. Spreading of viscous fluid drops on a solid substrate assisted by thermal fluctuations. *Phys. Rev. Lett.* **95**, 244505 (2005).
- [15] Mecke, K. & Rauscher, M. On thermal fluctuations in thin film flow. *J. Phys.: Condens. Matter* **17**, S3515–S3522 (2005).

- [16] Grün, G., Mecke, K. & M., R. Thin-film flow influenced by thermal noise. *J. Stat. Phys.* **122**, 1261–1291 (2006).
- [17] Zhang, Y., Sprittles, J. E. & Lockerby, D. A. Molecular simulation of thin liquid films: Thermal fluctuations and instability. *Phys. Rev. E* **100**, 023108 (2019).
- [18] Durán-Olivencia, M., Gvalani, R., Kalliadasis, S. & Pavliotis, G. A. Instability, rupture and fluctuations in thin liquid films: Theory and computations. *J. Stat. Phys.* **174**, 579–604 (2019).
- [19] Archer, A. J. & Rauscher, M. Dynamical density functional theory for interacting brownian particles: stochastic or deterministic? *J. Phys. A* **37**, 9325–9333 (2004).
- [20] Weeks, J. D. The roughening transition. In T., R. (ed.) *Ordering in Strongly Fluctuating Condensed Matter Systems*, 293–317 (Plenum, New York, 1980).
- [21] Dietrich, S., Nightingale, M. P. & Schick, M. Applicability of mean-field theory to wetting in binary-liquid mixtures. *Phys. Rev. B* **32**, 3182–3185 (1985).
- [22] Asakawa, H., Sazaki, G., Nagashima, K., Nakatsubo, S. & Furukawa, Y. Two types of quasi-liquid layers on ice crystals are formed kinetically. *Proc. Natl. Acad. Sci. U.S.A.* **113**, 1749–1753 (2016).
- [23] Murata, K.-i., Asakawa, H., Nagashima, K., Furukawa, Y. & Sazaki, G. Thermodynamic origin of surface melting on ice crystals. *Proc. Natl. Acad. Sci. U.S.A.* **113**, E6741–E6748 (2016).
- [24] Lutsko, J. F. A dynamical theory of nucleation for colloids and macromolecules. *J. Chem. Phys.* **136**, 034509 (2012).
- [25] Lutsko, J. F. How crystals form: A theory of nucleation pathways. *Sci. Adv.* **5**, eaav7399 (2019).
- [26] Marconi, U. M. B. & Tarazona, P. Dynamic density functional theory of fluids. *J. Chem. Phys.* **110**, 8032–8044 (1999).
- [27] Nenow, D. & Trayanov, A. Thermodynamics of crystal surfaces with quasi-liquid layer. *J. Cryst. Growth* **79**, 801–805 (1986).
- [28] Dash, J. G., Rempel, A. W. & Wettlaufer, J. S. The physics of premelted ice and its geophysical consequences. *Rev. Mod. Phys.* **78**, 695–741 (2006).
- [29] Wettlaufer, J. S. Surface phase transitions in ice: From fundamental interactions to applications. *Phil. Trans. R. Soc. A. Math. Phys. Eng. Sci.* **377**, 20180261 (2019).
- [30] Müller, M. & MacDowell, L. G. Interface and surface properties of polymer solutions: Monte Carlo simulations and self-consistent field theory. *Macromolecules* **33**, 3902–3923 (2000).

- [31] MacDowell, L. G. & Müller, M. Observation of autophobic dewetting on polymer brushes from computer simulation. *J. Phys.: Condens. Matter* **17**, S3523–S3528 (2005).
- [32] MacDowell, L. G. & Müller, M. Adsorption of polymers on a brush: Tuning the order of the wetting transition. *J. Chem. Phys.* **124**, 084907 (2006).
- [33] MacDowell, L. G. Computer simulation of interface properties: Towards a first principle description of complex interfaces? *Euro. Phys. J. ST* **197**, 131–145 (2011).
- [34] de Gregorio, R., Benet, J., Katcho, N. A., Blas, F. J. & MacDowell, L. G. Semi-infinite boundary conditions for the simulation of interfaces: The Ar/CO₂(s) model revisited. *J. Chem. Phys.* **136**, 104703 (2012).
- [35] MacDowell, L. G., Benet, J., Katcho, N. A. & Palanco, J. M. Disjoining pressure and the film-height-dependent surface tension of thin liquid films: New insight from capillary wave fluctuations. *Adv. Colloid Interface Sci.* **206**, 150–171 (2014).
- [36] Benet, J., Palanco, J. G., Sanz, E. & MacDowell, L. G. Disjoining pressure, healing distance, and film height dependent surface tension of thin wetting films. *J. Phys. Chem. C* **118**, 22079–22089 (2014).
- [37] MacDowell, L. G., Llombart, P., Benet, J., Palanco, J. G. & Guerrero-Martinez, A. Nanocapillarity and liquid bridge-mediated force between colloidal nanoparticles. *ACS Omega* **3**, 112–123 (2018).
- [38] Slater, B. & Michaelides, A. Surface premelting of water ice. *Nat. Rev. Chem* **3**, 172–188 (2019).
- [39] Chernov, A. A. & Mikheev, L. V. Wetting of solid surfaces by a structured simple liquid: Effect of fluctuations. *Phys. Rev. Lett.* **60**, 2488–2491 (1988).
- [40] Evans, R. Density functionals in the theory of nonuniform fluids. In Henderson, D. (ed.) *Fundamentals of Inhomogeneous Fluids*, chap. 3, 85–175 (Marcel Dekker, New York, 1992).
- [41] Henderson, J. R. Wetting phenomena and the decay of correlations at fluid interfaces. *Phys. Rev. E* **50**, 4836–4846 (1994).
- [42] Hughes, A. P., Thiele, U. & Archer, A. J. Influence of the fluid structure on the binding potential: Comparing liquid drop profiles from density functional theory with results from mesoscopic theory. *J. Chem. Phys.* **146**, 064705 (2017).
- [43] Oron, A., Davis, S. H. & Bankoff, S. G. Long-scale evolution of thin liquid films. *Rev. Mod. Phys.* **69**, 931–980 (1997).
- [44] Craster, R. V. & Matar, O. K. Dynamics and stability of thin liquid films. *Rev. Mod. Phys.* **81**, 1131–1198 (2009).
- [45] Thiele, U. Recent advances in and future challenges for mesoscopic hydrodynamic modelling of complex wetting. *Colloids. Surf. A* **553**, 487 – 495 (2018).

Reply to Reviewers

We thank again the reviewers for their revision of our manuscript and the lengthy supplementary materials section. We truly believe the article has considerably improved through the revision process.

A detailed response to the referees comments and a list of changes is found below.

Reply to Reviewer #1

The authors fully addressed my concerns of the article. The paper was already very sound and novative, but its density made it very hard to embrace fully. Thanks to the significant amendments of the authors, I am convinced the article can reach a broader community, and it is now easier to grasp the context and target of the paper, as well a extract relevant information.

Therefore, I fully recommend the nice piece of work for publication in nature Communications.

We thank again Prof. Demange for the comments. We have very rarely found such a careful revision. We also believe the paper is now more transparent and will reach a broader audience.

Reviewer #2

Reviewer #2 does not appear to have replied back, but we feel his comments were very valuable and have allowed us to substantially improve our understanding of the stochastic nature of the partial differential equations involved in this work.

Reviewer #3

I think the paper has been substantially improved through the revisions all the referees have suggested and I think the paper can be accepted now. The reply to my question has not been dealt with fully. The only way to assess the influence of dipoles is by performing a simulation with a more sophisticated water model such as that due to Paesani et al. but I am not confident such a simulation is tractable and in any case, publication of the study now may encourage the field to strive to test how robust the conclusions from the fixed dipole model are

We thank Reviewer#3 for the comments. We would be very happy if our paper could stimulate further studies and validation with a quantum mechanical calculation or many body potential. We realize that our previous response might not have been addressed this issue sufficiently and have added a new paragraph in section 'Interface potential for water on ice' to discuss this further. Two new references, Ref.42 and Ref.43 have also been added. Since we were already at the reference number limit, previous references 67 and 70 have been moved to the Supplementary Materials section.